# From Softmax to Score: Transformers Can Effectively Implement In-Context Denoising Steps

**Paul Rosu** [*]
Duke University
paul.rosu@duke.edu

**Lawrence Carin**
Duke University
lcarin@duke.edu

**Xiang Cheng**[*]
Duke University
xiang.cheng@duke.edu

## Abstract

Transformers have emerged as powerful meta-learners, with growing evidence that they implement learning algorithms within their forward pass. We study this phenomenon in the context of denoising, presenting a unified framework that shows Transformers can implement (a) manifold denoising via Laplacian flows, (b) score-based denoising from diffusion models, and (c) a generalized form of anisotropic diffusion denoising. Our theory establishes exact equivalence between Transformer attention updates and these algorithms. Empirically, we validate these findings on image denoising tasks, showing that even simple Transformers can perform robust denoising both with and without context. These results illustrate the Transformer's flexibility as a denoising meta-learner. Code available at `https://github.com/paulrosu11/Transformers_are_Diffusion_Denoisers`

## 1 Introduction

Two dominant paradigms in generative modeling—diffusion models and Transformer-based sequence models—have independently achieved remarkable success. Denoising diffusion probabilistic models (DDPMs) generate data by reversing a stochastic noise process, reaching state-of-the-art results in image synthesis [14, 2]. Transformers, trained autoregressively, power large language models (LLMs) that excel at text generation [15, 16]. This convergence prompts a natural question: can attention mechanisms perform *diffusion-like denoising*, and what does this imply about the nature of generative modeling?

Recent work shows these paradigms are deeply connected. Diffusion Transformers are competitive with U-Nets as backbones of diffusion models [14, 2], while diffusion-based language models like Diffusion-LM [12] offer controllable text generation, suggesting attention can drive iterative denoising in both vision and language. Beyond empirical advances, Transformers have demonstrated remarkable capabilities in meta-learning, particularly through in-context learning, where they emulate learning algorithms during inference without explicit parameter updates [3, 4]. Recent studies show that Transformers can approximate gradient-based updates in their forward pass [7, 1], effectively bridging meta-learning and classical optimization [6]. From a theoretical standpoint, Transformer dynamics have recently been linked to continuous-time probability flows and Wasserstein gradient flows [8], offering a principled lens for understanding attention-driven generation. Yet our theoretical understanding of Transformers as denoisers remains limited. How could attention layers collectively implement a stepwise *denoising* process—a generative *flow* through high-dimensional space? While invertible attention has been proposed in flow-based models [19], a general framework for *attention as a diffusion operator* is still lacking.

We seek to develop theoretical insight into how Transformers can perform iterative denoising, by analyzing the capability of Transformers to implement various denoising algorithms, and drawing

---

[*]Correspondence: `paul.rosu@duke.edu`, `xiang.cheng@duke.edu`

39th Conference on Neural Information Processing Systems (NeurIPS 2025).

connections to their meta-learning behavior in denoising contexts. Understanding attention as a generative flow could unify the strengths of both paradigms, and illuminate why Transformers are such powerful generative learners.

Denoising is central to generative modeling, yet theory on in-context learning (ICL) has largely focused on supervised settings with clean labels. Real contexts are noisy or partially observed, motivating unsupervised ICL. We ask whether Transformers can perform stepwise denoising during inference, and show that attention naturally implements structured denoising updates and, when trained end-to-end, can learn geometry-aware anisotropic diffusion. This yields a unified view of score-based diffusion and manifold learning as Transformer-implemented denoising and connects to recent in-context diffusion generation (e.g., [20]).

### 1.1 Outline and Main Contributions

The main contributions of our work are as follows:

1. **Transformers Implementing Manifold Denoising.** In Section 3, we consider the problem of **in-context manifold denoising** [10]: One is presented a set of points sampled from an arbitrary, context-dependent manifold perturbed by Gaussian noise, and the goal is to denoise these points. We show in Lemma 3 that *the Transformer can implement a Laplacian-based manifold-denoising algorithm* that iteratively applies a kernel-weighted update [10]. This result provides a clear theoretical link between manifold denoising and the attention mechanism of Transformers, highlighting their potential in learning implicit manifold structures in-context.

2. **Transformers as Exact Score-Based Diffusion Denoisers.** Section 4 examines the role of Transformers in the context of score-based diffusion generative models [11, 17]. We show rigorously that the Transformer architecture can **implement the exact score-based denoising algorithm** through a suitable cross-attention construction (Lemma 4). In Section 4.2, we further explore the Transformer's performance in an in-context denoising setup, demonstrating empirically that it effectively **generalizes from context-free to context-dependent denoising scenarios**, thus unifying these two important perspectives through the same attention-based mechanism.

3. **Efficient Approximate Score-Based Denoising via Learnable Witness Tokens.** Recognizing the impracticality of exact score-based denoising, Section 4.3 introduces an approach utilizing a small set of learnable tokens ("witnesses") that approximate the exact score computation. This significantly improves both accuracy and computational efficiency, as demonstrated in our empirical results in Section 4.3. The use of learnable witnesses bears a connection to several established methods in kernel approximation.

4. **Generalization to Anisotropic Diffusion and Learned Attention Parameters.** In Section 4.4, we generalize beyond isotropic kernels and standard diffusion models by studying **anisotropic diffusion processes**. We theoretically prove (Lemma 5) that Transformers, when their Query, Key, and Value parameters are suitably aligned with the diffusion coefficient matrix, can exactly implement the reverse ODE of anisotropic diffusion. This theoretical insight is substantiated empirically in Section 4.5, demonstrating that the Transformer can learn more **efficient, geometry-adaptive denoising algorithms beyond the standard score-based denoising framework**. We visualize this process in Figure 1.

Collectively, these contributions elucidate how attention and cross-attention can naturally realize a broad spectrum of denoising algorithms. Our results reveal the remarkable flexibility and effectiveness of Transformers for both context-dependent and context-independent denoising.

## 2 Transformer and Kernel Weighted Update

Given $n$ tokens $z^{(1)}, \ldots, z^{(n)} \in \mathbb{R}^d$, we define the matrix $Z_0 := \left[ z^{(1)}, \ldots, z^{(n)} \right] \in \mathbb{R}^{d \times n}$. For a $L$-layer Transformer, we let $W_\ell^V, W_\ell^Q, W_\ell^K \in \mathbb{R}^{d \times d}$ denote the value, query and key parameter matrices of layer $\ell$, and let $W_\ell^S \in \mathbb{R}^{d \times d}$ parameterize a linear module. For convenience, let $W_\ell := \left\{ W_\ell^V, W_\ell^Q, W_\ell^K, W_\ell^S \right\}$, and let $W := \{W_\ell\}_{\ell=0,\ldots,L}$. $\texttt{Attn}_{\texttt{std}}$ denotes the standard attention:

$$\texttt{Attn}_{\texttt{std}}\left(Z; W_\ell\right) := W_\ell^V Z \, \texttt{smax}\left( Z^\top W_\ell^{K^\top} W_\ell^Q Z \right), \tag{1}$$

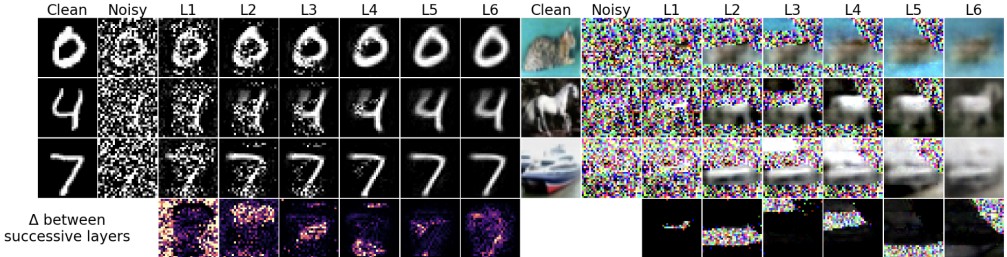

Figure 1: Visualization of diffusion steps of a denoising algorithm implemented by a *trained* Transformer with diagonal $W^V, W^Q, W^K$ matrices. The Transformer is modelled by the score-based anisotropic diffusion reversal algorithm in Lemma 5, Section 4.5. The first 2 columns show the clean and noisy images. Each subsequent column shows the output at a given layer of a 6-layer Transformer. We visualize the difference between consecutive layers' output at the bottom. Notice that each Transformer layer denoises a roughly non-overlapping patch.

where `smax` applies column-wise softmax. We use a diagonal mask so that each token does not attend to itself. This is standard when disallowing self-attention (diagonal mask) and is also common in manifold denoising, where the update at each position is informed by its neighbors rather than itself (see [10], Section 3.1). For notational simplicity, we omit the explicit mask matrix. Let $\texttt{rbf}(\cdot, \cdot) : \mathbb{R}^{d \times n} \times \mathbb{R}^{d \times n} \to \mathbb{R}^{n \times n}$ be the matrix valued RBF kernel, defined as $[\texttt{rbf}(U, V)]_{ij} = \exp\left(-1/2 \cdot \|U^{(i)} - V^{(j)}\|_2^2\right)$. Let $[\texttt{logrbf}(U, V)]_{ij} := [\log(\texttt{rbf}(U, V))]_{ij}$. We define $\texttt{Attn}_{\texttt{rbf}}$ as

$$\texttt{Attn}_{\texttt{rbf}}(Z; W_\ell) := W_\ell^V Z \, \texttt{smax}(\texttt{logrbf}(W_\ell^K Z, W_\ell^Q Z)). \tag{2}$$

We highlight that the difference between the standard $\texttt{Attn}_{\texttt{std}}$ and $\texttt{Attn}_{\texttt{rbf}}$ as follows:

$$[\texttt{smax}\left(Z^\top W_\ell^{K^\top} W_\ell^Q Z\right)]_{ij} \propto \exp\left((W_\ell^K z^{(i)})^\top (W_\ell^Q z^{(j)})\right)$$

$$[\texttt{smax}(\texttt{logrbf}(W_\ell^K Z, W_\ell^Q Z))]_{ij} \propto \exp\left((W_\ell^K z^{(i)})^\top (W_\ell^Q z^{(j)}) - \|W_\ell^K z^{(i)}\|_2^2/2 - \|W_\ell^Q z^{(j)}\|_2^2/2\right).$$

Given $\texttt{Attn} \in \{\texttt{Attn}_{\texttt{std}}, \texttt{Attn}_{\texttt{rbf}}\}$, we define the Transformer via the iterative update:

$$Z_{\ell+1} = W_\ell^S Z_\ell + \texttt{Attn}\left(Z_\ell; W_\ell^V, W_\ell^Q, W_\ell^K\right). \tag{3}$$

In many subsequent applications of interest, $\|W_\ell^Q z^{(j)}\|_2^2$ and $\|W_\ell^K z^{(j)}\|_2^2$ are constant, or close to constant, due to concentration of norms of high-dimensional Gaussian vectors. In this case, we verify that $\texttt{Attn}_{\texttt{std}}$ and $\texttt{Attn}_{\texttt{rbf}}$ behave identically:

**Lemma 1** (Equivalence of $\texttt{Attn}_{\texttt{std}}$ and $\texttt{Attn}_{\texttt{rbf}}$ on the sphere (with $W^Q, W^K$ identity-scaled)). *If $\|z_\ell^{(i)}\|_2 = C$ for some constant $C$, and if $W_\ell^Q$ and $W_\ell^K$ are scalings of the identity matrix, then $Attn_{std}(Z_\ell, W) = Attn_{rbf}(Z_\ell, W)$.*

We defer proof of Lemma 1 to Appendix A. We also empirically verify the similar performance of $\texttt{Attn}_{\texttt{std}}$ and $\texttt{Attn}_{\texttt{rbf}}$ in Figure 3.

## 2.1 Transformer Construction for Kernel-Weighted Update.

Given tokens $\{z^{(i)}\}_{i=1,\dots,n} \subset \mathbb{R}^d$, let $\mathcal{F} = \{z^{(1)}, \dots, z^{(m)}\}$ denote the subset of frozen tokens. Much of our discussion in this paper revolves around the following *kernel-weighted-update* scheme:

$$z_{\ell+1}^{(j)} = (1 - \alpha_\ell) z_\ell^{(j)} + \gamma_\ell \sum_{i \neq j} \kappa_\ell(i, j)\left(z_\ell^{(i)} - z_\ell^{(j)}\right) \qquad \text{for} \qquad j \notin \mathcal{F}, \tag{4}$$

$$\kappa_\ell(i, j) := \frac{\mathbb{1}\{i \neq j\} \exp(-\|z_\ell^{(i)} - z_\ell^{(j)}\|_2^2/(2\sigma_\ell^2))}{\sum_{k \neq j} \exp(-\|z_\ell^{(k)} - z_\ell^{(j)}\|_2^2/(2\sigma_\ell^2))} \tag{5}$$

and $z_{\ell+1}^{(j)} = z_\ell^{(j)}$ for $j \in \mathcal{F}$. Motivated by (4), we implement this update using a single attention module together with a fixed mask that freezes the columns in $\mathcal{F}$.

Let $Z_\ell := [z_\ell^{(1)}, \dots, z_\ell^{(n)}] \in \mathbb{R}^{d \times n}$, and let $M_\mathcal{F} \in \mathbb{R}^{n \times n}$ denote the diagonal *freeze mask* with $(M_\mathcal{F})_{jj} := \mathbb{1}\{j \notin \mathcal{F}\}$. We define the Transformer with frozen-token mask by

$$Z_{\ell+1} = \left( W_\ell^S Z_\ell + \texttt{Attn}_{\texttt{rbf}}\left( Z_\ell; W_\ell^V, W_\ell^Q, W_\ell^K \right) \right) M_\mathcal{F} + Z_\ell (I_{n \times n} - M_\mathcal{F}). \tag{6}$$

We verify below that (6) can exactly implement the kernel-weighted update (4) (proof deferred to Appendix A):

**Lemma 2** (Transformer implements kernel-weighted update.)**.** *By choosing* $W_\ell^Q = 1/\sigma_\ell I_{d \times d}, W_\ell^K = 1/\sigma_\ell I_{d \times d}, W_\ell^V = \gamma_\ell I_{d \times d}, W_\ell^S = (1 - \alpha_\ell - \gamma_\ell) I_{d \times d}$, *the masked Transformer in* (6) *exactly implements the kernel-weighted update in* (4)*.*

**Cross-attention form.** When a subset of tokens is permanently frozen (e.g., a large training set), it is convenient to store them separately and update only the remaining tokens. Let $Z^\mathcal{F} := [z^{(1)}, z^{(2)}, \dots, z^{(m)}] \in \mathbb{R}^{d \times m}$ denote the matrix whose columns are the frozen tokens. Let $Z := [z^{(m+1)}, z^{(m+2)}, \dots, z^{(n)}]$ denote the matrix of not-frozen tokens. Let cross attention modules $\texttt{CrAttn}_{\texttt{std}}(U, Z; W_\ell) := W_\ell^V U \texttt{smax}\left( U^\top W_\ell^{K^\top} W_\ell^Q Z \right)$ and $\texttt{CrAttn}_{\texttt{rbf}}(U, Z; W_\ell) := W_\ell^V U \texttt{smax}(\texttt{logrbf}(W_\ell^K U, W_\ell^Q Z))$ be defined analogously to $\texttt{Attn}_{\texttt{std}}$ and $\texttt{Attn}_{\texttt{rbf}}$. We consider the Transformer with cross attention:

$$Z_{\ell+1} = W_\ell^S Z_\ell + \texttt{Attn}_{\texttt{rbf}}\left( Z_\ell; W_\ell^V, W_\ell^Q, W_\ell^K \right) + \texttt{CrAttn}_{\texttt{rbf}}\left( Z^\mathcal{F}, Z_\ell; W_\ell^{V'}, W_\ell^{Q'}, W_\ell^{K'} \right). \tag{7}$$

We will use $\texttt{TF}_\ell(Z, Z^\mathcal{F}; W)$ to denote the value of $Z_\ell$, when evolved according to (7), initialized at $Z_0 = Z$ and $Z^\mathcal{F}$; i.e. the output of the $\ell$-layer Transformer in (7). In the single-query setting used for score-based denoising (Section 4), (7) coincides with (6) (the self-attention term vanishes), so we will use the cross-attention notation for convenience.

## 3 Manifold Denoising via Laplacian-based ODE

In this section, we are motivated by the following *in-context manifold denoising* problem, originally proposed in [10]: Let $\mathcal{M}$ be a $m$-dimensional manifold embedded via an embedding $T : \mathcal{M} \to \mathbb{R}^d$. We identify a point $a \in \mathcal{M}$ with its $d$-dimensional embedding $T(a)$. Let $x^{(1)}, \dots, x^{(n)}$ be sampled from $\mathcal{M}$, and we observe $z^{(i)} = x^{(i)} + \varepsilon^{(i)}$, where $\varepsilon^{(i)} \sim \mathcal{N}(0, \sigma^2 I_{d \times d})$. The goal is to recover the $x^{(i)}$'s. This problem is particularly challenging because (1) the manifold $\mathcal{M}$ is unobserved, and (2) the manifold $\mathcal{M}$ is *context-dependent*. Therefore, to solve this problem *in-context*, the Transformer needs to be able to learn (implicitly or explicitly) the manifold structure, from the observations $z^{(i)}$'s. [10] propose the following manifold-denoising algorithm based on the Laplace-Beltrami operator: Let the adjacency matrix $\mathcal{W} \in \mathbb{R}^{n \times n}$ be defined as $\mathcal{W}_{ij} := \exp(-\|z^{(i)} - z^{(j)}\|_2^2/(2\sigma^2))$ and set $\mathcal{W}_{ii} = 0$ to remove self-loops. Let $\mathcal{D}$ be the diagonal matrix with $\mathcal{D}_{ii} := \sum_{j=1}^n \mathcal{W}_{ij}$. The RBF Laplacian is defined as $\mathcal{L} = I_{n \times n} - \mathcal{D}^{-1} \mathcal{W}$. The denoising algorithm is based on a Laplacian-based flow $\partial_t z^{(i)}(t) = -\gamma \sum_{j=1}^n \mathcal{L}_{ij} z^{(j)}(t)$, whose Euler discretization gives the discrete-time algorithm:

$$z_0^{(i)} = z^{(i)}, \qquad z_{\ell+1}^{(i)} = z_\ell^{(i)} - \delta_\ell \Big( \sum_{j=1}^n \mathcal{L}_{ij} z_\ell^{(j)} \Big). \tag{8}$$

In Lemma 3, we verify that (8) is an instance of the kernel-weighted update (4) (with $\mathcal{F} = \varnothing$), and is therefore implemented by the masked-attention Transformer construction in Lemma 2.

**Lemma 3** (Reformulation of Manifold Denoising)**.** *The manifold denoising algorithm in* (8) *is exactly equivalent to* (4)*, with* $\mathcal{F} = \varnothing$*,* $\alpha_\ell = 0$*,* $\sigma_\ell = \sigma$*, and* $\gamma_\ell = \delta_\ell$*. Consequently, the algorithm* (8) *is implemented by the Transformer construction in Lemma 2.*

The proof of Lemma 3 is just by algebra; we defer it to Appendix B.

## 3.1 Experiments for Manifold Denoising

We present experimental evidence demonstrating the Transformer construction in Lemma 2 can indeed act as an effective manifold denoiser. The problem setup is very similar to the above: Let $x^{(1)}, \ldots, x^{(n)}$ be clean images sampled from a training set. We perturb each image via $z^{(i)} = x^{(i)} + \varepsilon^{(i)}$, where $\varepsilon^{(i)} \sim \mathcal{N}(0, \sigma^2 I)$, for some noise level $\sigma$. The input matrix $Z_0 = [z^{(1)}, \ldots, z^{(n)}]$ and the Transformer $\mathsf{TF}_\ell(Z; W_\ell)$ are as defined in Section 2. Motivated by Lemma 2, we impose the constraint that $W_\ell^V, W_\ell^Q, W_\ell^K, W_\ell^S$ are each a (trainable) scalar multiple of identity. The training loss is given by $\mathbb{E}_{Z_0}\left[\frac{1}{n}\sum_{i=1}^n \|[\mathsf{TF}_L(Z_0; W)]^{(i)} - x^{(i)}\|_2^2\right]$. Throughout, we report noise levels via a signal-to-noise ratio (SNR) defined as $\mathrm{SNR} := \sigma_{\mathrm{noise}}/\sigma_{\mathrm{data}}$, i.e., the standard deviation of the additive Gaussian noise divided by that of the clean data distribution.

Figure 2a shows that the test loss decreases with the context length; this agrees with the fact that the discrete RBF Laplacian $\mathcal{L}$ becomes a better approximation to the Laplace-Beltrami operator with increasing number of samples. Figure 2b shows the test loss of the intermediate layer outputs of a 6-layer Transformer; the test loss decreases with number of layers, which is consistent with (8), as each layer implements one more discretization step of (8).

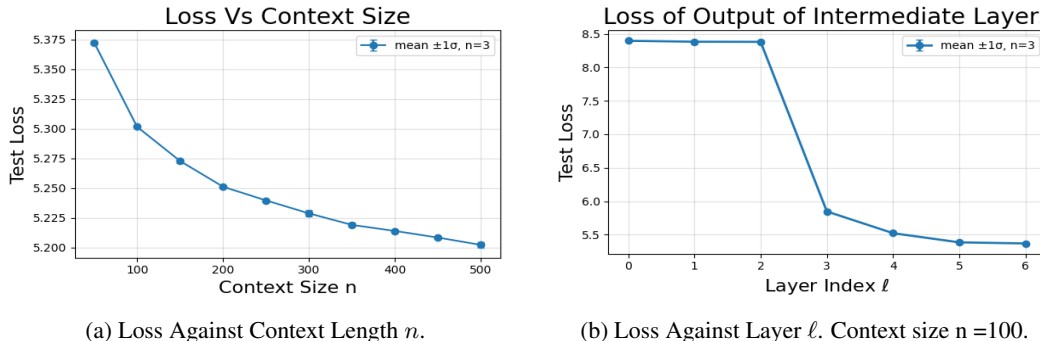

(a) Loss Against Context Length $n$.    (b) Loss Against Layer $\ell$. Context size n =100.

Figure 2: (a) Test-set RMSE on MNIST versus context size $n$ for the Laplacian-based denoising Transformer under a noise level of SNR $\approx 1$. Each point shows the converged loss of a separately trained 6-layer model; (b): Test RMSE of intermediate output at layer $\ell$ with fixed $n = 100$.

## 4 Score-Based Diffusion Denoising

As in standard score-based diffusion models [11, 18, 17], our training objective is defined over the empirical distribution $\hat{p}$ (equivalently denoted $p_0$ below), the uniform distribution over training samples $\{x^{(i)}\}_{i=1}^n$. We assume these samples are drawn i.i.d. from an unknown population distribution $p$. While optimization is with respect to $\hat{p}$, the modeling goal is to generalize to $p$. Unless otherwise stated, all evaluations reported in this section are computed on unseen test examples.

Score-based diffusion generative models [11, 18] motivate the following problem: Let $p_0$ denote the uniform distribution over a (large) training set of samples $\{x^{(1)}, \ldots, x^{(n)}\}$. Let $p_t = p_0 * \mathcal{N}(0, tI)$, where $*$ denotes convolution. As we review in Section 4.1, getting access to $\nabla \log p_t(x)$ allows one to turn an (almost pure) Gaussian noise $x$ into a sample from $p_0$ by reversing a diffusion process. In practice, one trains a neural network $s_\theta(x, t)$, parameterized by weights $\theta$, to approximate the (scaled) score, by minimizing the denoising objective $\mathbb{E}\left[\|s_\theta(x; t) - t\nabla \log p_t(x)\|_2^2\right]$.

Conveniently, $\nabla \log p_t(x)$ is exactly equal to a weighted-average between $x$ and each $\{x^{(i)}\}_{i=1,\ldots,n}$. Consequently, we verify in Lemma 4 that the Transformer (7) can implement score-based denoising.

### 4.1 Standard Score-based Diffusion Denoising

Consider the variance-exploding SDE, defined as

$$x_0 \sim p_0, \qquad dx_t = dB_t, \tag{9}$$

where $B_t$ denotes the standard Brownian motion. Let $p_t$ denote the distribution of $x_t$. We verify that $p_t = p_0 * \mathcal{N}(0, tI)$. By the Fokker Planck equation, $p_t$ evolves as

$$v_t(x) := -\frac{1}{2} \nabla \log p_t(x) \qquad \Rightarrow \qquad \frac{\partial}{\partial t} p_t(x) = -\mathbf{div}(p_t(x) v_t(x)). \tag{10}$$

Consequently, if $z_0 \sim p_0$ and $dz_t = -\frac{1}{2} \nabla \log p_t(z_t) dt$, then $z_t$ has the same distribution as $x_t$ in (9). The above follows directly from integration-by-parts. The diffusion process of (9) can be reversed by reversing the velocity field $v_t(x)$. Concretely, for any fixed time $T$, let $y_0 \sim p_T$, and $dy_s = \frac{1}{2} \nabla \log p_{T-s}(y_s) ds$, then $y_s \sim p_{T-s}$. The Euler discretization of the ODE for $y_s$ is given by

$$\sigma_{\ell+1}^2 = \sigma_\ell^2 - \delta_\ell, \qquad \bar{y}_{\ell+1} = \bar{y}_\ell + \frac{\delta_\ell}{2} \nabla \log p_{\sigma_\ell^2}(\bar{y}_\ell), \tag{11}$$

where $\sigma_k^2$ is the time $T - s$ at step $k$ (equivalently, the Gaussian variance of $\bar{y}_\ell$).

From the definition of $p_0$ and $p_t$, it follows that

$$p_t(x) = \frac{1}{nZ_t} \sum_{i=1}^n e^{-\frac{1}{2t} \left\| x - x^{(i)} \right\|_2^2} \quad \Rightarrow \quad \nabla \log p_t(x) = \frac{\sum_{i=1}^n e^{-\frac{1}{2t} \left\| x - x^{(i)} \right\|_2^2} \left( x^{(i)} - x \right)}{t \sum_{i=1}^n e^{-\frac{1}{2t} \left\| x - x^{(i)} \right\|_2^2}}, \tag{12}$$

where $Z_t$ is the normalization constant for $\mathcal{N}(0, tI)$. Observe that the RHS of (12) is exactly the kernel-weighted average $\kappa$ from (4). Combining (12) and (11) then implies that the score-based denoising algorithm (11) is equivalent to (4), which we formalize in the following lemma:

**Lemma 4** (Score-based Denoising as Kernel-Weighted Update)**.** *The score-based denoising algorithm* (11) *is equivalent to* (4) *with* $z_\ell^{(n+1)} := \bar{y}_\ell$, $\mathcal{F} = \{1, \ldots, n\}$, $\alpha_\ell = 0$, *and* $\gamma_\ell = \delta_\ell / (2\sigma_\ell^2)$. *Consequently, the Transformer construction in Lemma 2 can implement score-based denoising* (11).

We defer the proof to Appendix C. We highlight an important difference from the manifold denoising setup in Section 3: here, the set of training samples is typically *large, and does not vary with context*. Instead, the training samples encode *global context-free knowledge*. On the other hand, the presence of similar tokens *in-context* can indeed improve the denoising accuracy; we show this empirically in Figure 5 in the next section. Thus **the same attention architecture can be well-suited for both in-context, and context-free learning**.

In Section 4.3, we discuss a generalization of the construction of Lemma 4; where we train the cross-attention tokens in (7) to learn an approximate score that significantly improves generalization error and computational cost. In Section 4.4, we discuss a generalization of the score-based denoising sequence (11) itself, which also leads to significant accuracy improvements.

## 4.2 Experiments for Score-Based Denoising

We empirically validate Lemma 4. Our experiment verifies that the Transformer construction of Lemma 4 does indeed implement exact score-based denoising. The setup is as follows: for each input context, we sample $x^{(1)}, \ldots, x^{(n)}$ clean images from a training set. We let $z^{(i)} = x^{(i)}$. The query image is $x^{(n+1)}$. We perturb the query image by Gaussian noise: $z^{(n+1)} = x^{(n+1)} + \varepsilon^{(n+1)}$, where $\varepsilon^{(n+1)} \sim \mathcal{N}(0, \sigma^2 I)$, where $\sigma$ is chosen noise level. The Transformer $\mathsf{TF}_\ell(Z; W_\ell)$ is as defined in (7). Motivated by Lemma 2, we impose the constraint that $W_\ell^V, W_\ell^Q, W_\ell^K, W_\ell^S$ are each a (trainable) scalar multiple of identity. The training loss is given by the reconstruction error $\mathbb{E}_{Z_0} \left[ \| [\mathsf{TF}_L(Z_0; W)]^{(n+1)} - x^{(n+1)} \|_2^2 \right]$.

**Single-query score-based denoising.** Figure 3 shows two architectures: {"RBF", "STD"} refer to Transformers that use $\mathtt{Attn_{rbf}}$ (2) and $\mathtt{Attn_{std}}$ (1) respectively. For each architecture, "trained" means the Transformer is trained on the above loss; "theory" means the Transformer parameters are fixed at the construction in Lemma 4 (for a geometrically decreasing $\sigma_k$ schedule). Figure 3 plots the test loss of the *per-layer output* of 6-layer Transformers. We observe the following:

1. The reconstruction error decreases monotonically with layers, suggesting that the Transformer is implementing an iterative denoising algorithm at training convergence. "RBF-trained" appears to denoise more aggressively in the first few layers compared to the theory constructions.

2. RBF-theory and STD-theory have almost identical losses, supporting our claim in Lemma 1 that the RBF and STD attentions behave similarly under our constructions.

In Figure 4, we visualize the clean and noisy query images, as well as the per-layer intermediate outputs of the RBF-trained and RBF-theory Transformers from Figure 3. In Figure 4a and 4b, we see that the trained Transformer can exactly recover the clean *training set image*, as does the exact score-denoising algorithm. In Figure 4c and 4d, we see that test images are not being recovered; instead the denoising algorithm produces a similar image from the training set. Though undesired, this is the expected behavior for exact score-denoising. In Figure 4e and 4f, we see the "generation" capabilities of each Transformer from pure noise; again, the "generated" image is in fact a sample from the training set (as expected of exact score denoising). The generated outputs of RBF-trained and RBF-theory are almost identical in all cases.

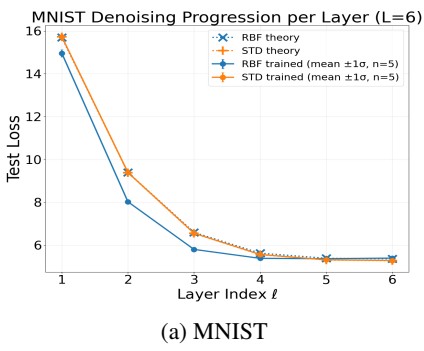 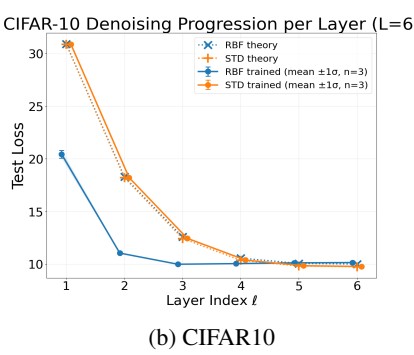

(a) MNIST            (b) CIFAR10

Figure 3: (a) Test loss of intermediate output at layer $\ell$ versus $\ell$ on MNIST, (b) CIFAR-10, at an injected noise level corresponding to SNR $\approx 3$ for (a) and $\approx 5$ for (b). All Transformers have 6-layers total. For CIFAR-100 results and FID on CIFAR-10/100, see App. E.2.1.

**In-context score-based denoising.** In Figures 5 and 6, we present a different experiment which *highlights the in-context aspect of score-based denoising.* In this setup, the input consists of the noiseless training samples $\{x^{(1)}, \ldots, x^{(n)}\}$, as well as $m$ **contextual images** $\{x^{(n+1)}, \ldots, x^{(n+m)}\}$. The query image is $x^{(n+m+1)}$. Crucially, all $m$ contextual images *belong to the same class as the query image $x^{(n+m+1)}$*. The Transformer input is $z^{(i)} = x^{(i)}$ for $i = 1, \ldots, n + m$, and the query point is perturbed with Gaussian noise: $z^{(n+m+1)} = x^{(n+m+1)} + \mathcal{N}(0, \sigma^2 I)$. We call $m$ the "context length". The query $x^{(m+n+1)}$ attends to the training set $x^{(1)}, \ldots, x^{(n)}$ via the `CrAttn` module in (7), while the context samples + query $x^{(n+1)}, \ldots, x^{(n+m)}$ attend to each other via the `Attn` module in (7) (with appropriate masking). Experiment details are in Appendix E.4.

The training loss is again the reconstruction loss. In Figure 5, we plot the test reconstruction loss against the context length. The presence of a few contextual examples leads to a significant improvement in reconstruction error. This demonstrates that the Transformer is indeed implementing a denoising algorithm that can adapt to its context (at least partially). Figure 6 visualizes this phenomenon. The injected noise has high SNR=5, so that the original image is difficult to recover. In the leftmost block, we see that providing same-class contextual images guided the model to correctly generate "6" (even though the intermediate output appears to tend towards "5"). In the middle block, the contextual set consisted of images from random classes, and the Transformer had difficulty recovering the image. In the right block, a Transformer trained without context generated a different digit from the same starting noisy image.

### 4.3 Approximate Score-Based Denoising With Learnable Witnesses

In the preceding section, we show that exact score-based denoising can be implemented via cross-attention between the noisy sample, and every other sample in the training dataset. This is in generally practically infeasible. On the flip-side, *the training set is typically not context-dependent.* Therefore, one can hope to approximate the exact score *cross-attending to a small set of representative samples*. Concretely, let $\Psi = \{\psi^{(1)}, \ldots, \psi^{(\tau)}\} \subset \mathbb{R}^d$ denote a set of *witnesses* $\psi^{(i)}$, for some

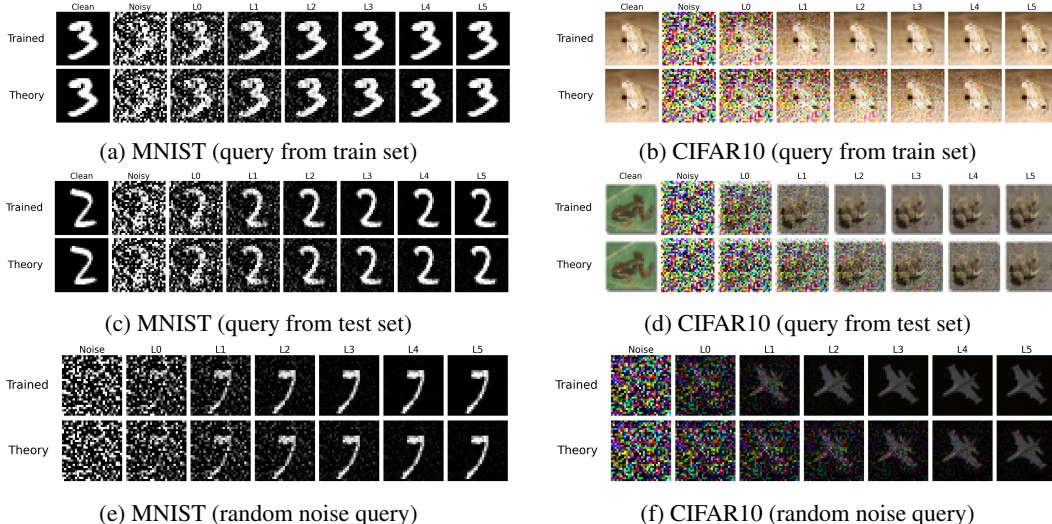

(a) MNIST (query from train set)  (b) CIFAR10 (query from train set)

(c) MNIST (query from test set)  (d) CIFAR10 (query from test set)

(e) MNIST (random noise query)  (f) CIFAR10 (random noise query)

Figure 4: Visualization of denoised images. Query image from {train, test, random noise}. In each sub-figure, the 1st row shows the trained Transformer, and the 2nd row shows the Transformer with theoretical weights. First two columns in each figure shows the {clean, noisy} query image respectively. Columns 2–7 show the per-layer intermediate outputs of a 6-layer Transformer.

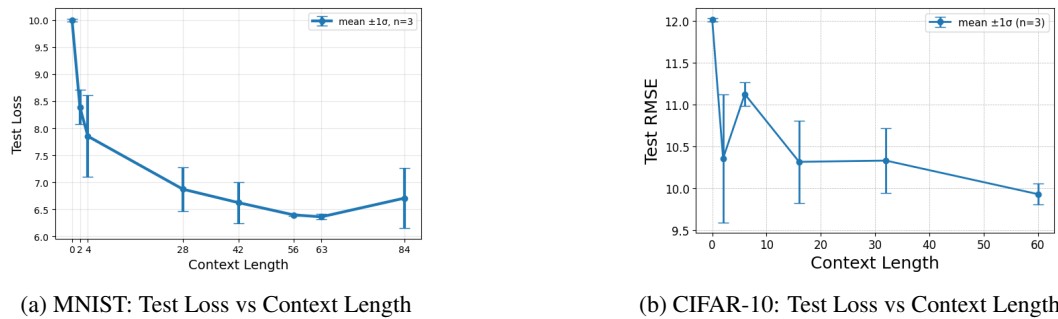

(a) MNIST: Test Loss vs Context Length  (b) CIFAR-10: Test Loss vs Context Length

Figure 5: Test-set RMSE for a 6-layer RBF-attention denoiser as a function of the number of clean, same-class context images $n_{\text{ctx}}$. The query token is severely noised (SNR $\approx 15$ for MNIST, SNR$\approx$ 10 for CIFAR-10), while the full training set is available through cross-attention. Adding just a few class-matched examples dramatically lowers the loss from the nearly unrecoverable zero-context case. For CIFAR-100 results and FID on CIFAR-10/100, see App. E.3

$\tau << n$. The goal is to choose a $\Psi$ which well-approximates the kernel-weighted update from (4).

$$\underbrace{\sum_{j=1}^{\tau} \frac{e^{-\|\psi^{(j)}-z\|_2^2/(2\sigma^2)}}{\sum_{k=1}^{\tau} e^{-\|\psi^{(k)}-z\|_2^2/(2\sigma^2)}} \left( \psi^{(j)} - z \right)}_{\text{①}} \approx \underbrace{\sum_{j=1}^{n} \frac{e^{-\|z^{(j)}-z\|_2^2/(2\sigma^2)}}{\sum_{k=1}^{n} e^{-\|z^{(k)}-z\|_2^2/(2\sigma^2)}} \left( z^{(k)} - z \right)}_{\text{②}}. \quad (13)$$

Observe that the ② is equivalent to the $\kappa$-averaged term in (4). In the setting of score-based denoising, $z^{(1)}, \ldots, z^{(n)}$ is drawn from $p_0$ independently. An intuitively simple choice of $\Psi$ is then to sub-sample $\tau$ points from $\{z^{(1)}, \ldots, z^{(n)}\}$; under this choice, the LHS of (13) is simply a Monte-Carlo estimate of the RHS of (13). Assume that $p_0$ is compactly supported, we can show via basic algebra that $\mathbb{E}\left[\|① - ②\|_2^2\right] = O(\tau^{-1})$. We provide a short proof in Lemma 7 in Appendix C. Much more sophisticated approaches exist, and the problem of picking a good witness set $\Psi$ has deep connections to a number of areas such as Nystrom's method for kernel approximation [21, 9], kernel mean embeddings [13] and kernel herding [5].

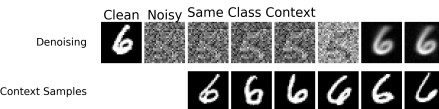 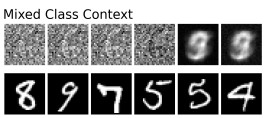 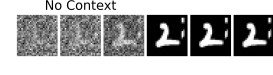

Figure 6: Two leftmost images are the clean and noisy (SNR $\approx 15$) query images respectively. We show three sets of six intermediate layer outputs. Left: a Transformer trained using with a 100-image contextual set is presented contextual images from the correct class at test time, and generates the correct image. Middle: the same Transformer, when provided contextual images from random classes, fails to generate the right image. Right: a different Transformer, trained without context, generates the digit 2 from the same input. All three scenarios start from the identical noisy input.

Motivated by the above, we treat $\Psi$ as a *learnable parameter*, alongside the other Transformer parameters. Lemma 4 showed that the score-based denoising algorithm (11) can be implemented by the Transformer defined in (6) (equivalently (7)). The witness-based Transformer is then defined as

$$Z_{\ell+1} = W_\ell^S Z_\ell + \texttt{Attn}_{\texttt{rbf}}\left(Z_\ell; W_\ell^V, W_\ell^Q, W_\ell^K\right) + \texttt{CrAttn}_{\texttt{rbf}}\left(U_\ell, Z_\ell; W_\ell^{V'}, W_\ell^{Q'}, W_\ell^{K'}\right),$$
(14)

where $U_\ell \in \mathbb{R}^{d \times \tau}$ is the only difference from (7). The full set of parameters of (14) is $\{U_\ell, W_\ell^V, W_\ell^Q, W_\ell^K\}_{\ell=1,\dots,L}$. In words, column $i$ of $U_\ell$ represents $\psi^{(i)}$ in the above discussion, and $U_\ell$ can differ from layer to layer (analogous to $\Psi$ changing with each denoising step.) We initialize $U_\ell$ with randomly sampled vectors from $\{z^{(i)}\}_{i=1,\dots,n}$. We show in Figure 7 that (14) has significant advantages in computation and generalization error (see Section 4.5 for discussion).

### 4.4 Anisotropic Diffusion and its Reverse ODE

Consider a generalization of the standard diffusion process in (9): instead of $dx_t = dB_t$, let $A_t : \mathbb{R}^+ \to \mathbb{R}^{d \times d}$ denote a time-parameterized family of matrices. For simplicity, we assume that $A_t$ is symmetric. The *anisotropic diffusion* is described by

$$x_0 \sim p_0, \qquad dx_t = A_t dB_t.$$
(15)

Throughout Section 4.4 we restrict to diffusion schedules satisfying the normalization $\int_0^T A_s^2 \, ds = TI$. Let $p_t$ denote the distribution of $x_t$. Let $T$ be some final time, and let $p_T = p_0 * \mathcal{N}(0, TI)$. In Appendix C.1, we show that (15) admits a simple continuous-time reverse ODE $y_t \sim p_{T-t}$. Its Euler discretization gives a denoising algorithm that can be implemented by a Transformer:

**Lemma 5** (Anisotropic Denoising with Transformer). *Let $x^{(1)}, \dots, x^{(n)}$ denote a training set. Let $z^{(i)} = x^{(i)}$ for $i = 1, \dots, n$. Let $z^{(n+1)} = x^{(n+1)} + \mathcal{N}(0, TI)$ denote the query token. Consider the anisotropic diffusion in (15), where $A_t$ is a family of arbitrary symmetric matrices satisfying $\int_0^T A_s^2 ds = TI$. Let $t_0 := T$, $t_{\ell+1} := t_\ell - \delta_\ell$, $M_0 := TI$, $M_{\ell+1} = M_\ell - \delta_\ell A_{t_\ell}^2$, where we identify $A_\ell$ with $A_{t_\ell}$. Then there exists a discrete-time denoising algorithm, based on the time-reversal ODE of the anisotropic diffusion in (15). This denoising algorithm can be implemented by the Transformer in (7), with $\mathcal{F} = \{1, \dots, n\}$ and parameters*

$$W_\ell^S = I - \delta_\ell A_\ell^2 M_\ell^{-1}/2, \quad W_\ell^V = \delta_\ell A_\ell^2 M_\ell^{-1}/2, \quad W_\ell^Q = W_\ell^K = M_\ell^{-1/2}.$$
(16)

The proof of Lemma 5 uses the same tools from Section 4.1, and we defer it to Appendix C. The importance of Lemma 5 lies in **generalizing beyond the isotropic RBF kernel** used in score-based denoising, and justifies the choice of **non-identity** $W^S, W^Q, W^K, W^V$ **matrices**, which enables the Transformer to implement an anisotropic denoising algorithm that adapts to the distribution at step $\ell$. In Section 4.5, we evaluate a witness-variant of the Transformer from Lemma 5.

### 4.5 Experiments for Witness and Anisotropic Denoising

In the following, we present a set of experiments that serves two purposes: First, it demonstrates the advantage of the Transformer with learnable witnesses from Section 4.3. Second, it studies the anisotropic diffusion Transformer construction from Lemma 5, with general learnable

$W^Q, W^K, W^V$ matrices. The setup is identical to the single-query score-based denoising experiment in in Section 3.1. The one difference is in the architecture: let the Transformer $\mathsf{TF}_\ell(z_0; U_\ell, W_\ell)$ be as defined in (14), where $U_\ell$ denotes the set of learnable witnesses and $W_\ell$ denotes the set of parameters at layer $\ell$. The training loss is given by $\mathbb{E}_{Z_0}\left[\|\mathsf{TF}_L(z; U, W) - x\|_2^2\right]$.

We initialize the witnesses (U) by sampling a random subset of training examples to set the initial columns of (U). During training, we treat (U) as a continuous, fully learnable parameter matrix and optimize it jointly with (W) via gradient descent. Consequently, at convergence the columns of (U) need not coincide with any training example (e.g., a column of (U) may represent an average or a denoising-friendly summary of multiple samples). This makes witness selection differentiable end-to-end and connects to classical kernel approximation methods (e.g., Nystrom and related constructions) that summarize large kernels with a small set of inducing points [21, 9, 5, 13].

In Figure 7, we compare the test loss of three models: the Baseline model is the Transformer from Lemma 4 that exactly implements the score-based denoising algorithm in (11). The "Witness+RBF" model implements the witness-Transformer from (14), and its weights $W$ are (trainable) scalar multiples of *identity*. The "Witness+anisotropic" model is motivated by Lemma 5; we do not assign the Transformer parameters as done in (16); instead, we simply make $\{W_\ell^S, W_\ell^V, W_\ell^Q, W_\ell^K\}$ trainable *diagonal* matrices. We provide details of the implementation in Appendix E.4.

With a moderate number of witnesses, both witness-based Transformers significantly outperform the baseline Transformer that implements exact score matching. Furthermore, the anisotropic denoising Transformer shows a significant advantage over the RBF (isotropic) denoising Transformer.

In Figure 1, we visualize the per-layer denoising progress of a 6-layer Witness+anisotropic Transformer (1000 witnesses). In contrast to the standard denoising sequence in Figure 4, where noise is removed uniformly in space, **the denoising sequence in Figure 1 proceeds patch-wise**. Such a patch-wise denoising procedure coincides with the anisotropic denoising algorithm when $A_\ell$ for each step $\ell$ is a sparse diagonal matrix, whose non-zero entries coincide with the pixel locations of a localized image patch.

Across datasets and witness counts, the witness-based Transformer achieves lower test reconstruction error than the Baseline model that implements exact score-based denoising. Although the Baseline can perfectly fit the empirical train distribution ($\hat{p}$), its unconstrained attention tends to generalize poorly. In contrast, the Witness model's learnable summary witnesses provide an inductive bias that improves generalization to the population distribution ($p$).

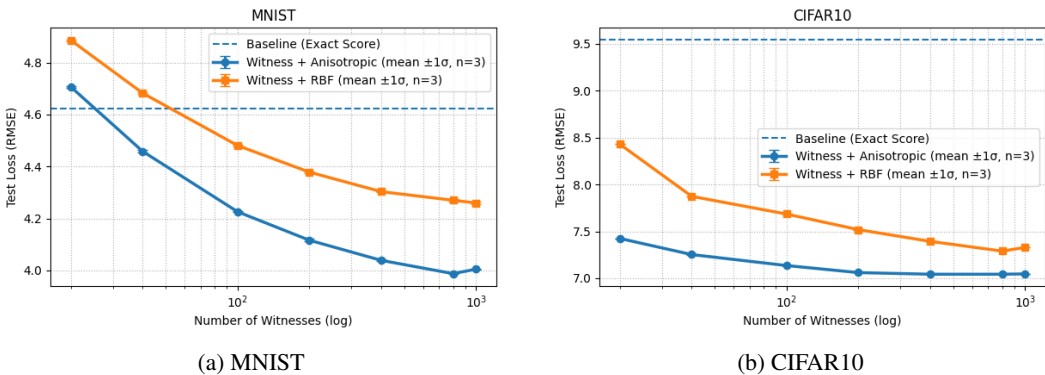

(a) MNIST  (b) CIFAR10

Figure 7: Test loss at SNR $\approx 3$ for (a) and $\approx 5$ for (b) with a 6-layer model as a function of the number of trainable witness tokens $\tau$. The horizontal line shows the performance of the exact-score denoiser that uses the full training set, while the two curves plot witness-based approximations (14): one based on anisotropic diffusion (Section 4.4), and one based on standard RBF kernel. All reported metrics are computed on unseen test data to assess generalization. For CIFAR-100 results and FID on CIFAR-10/100, see App. E.3.1.

All experiments are done on a single A5000 GPU, with each experiment taking at most a few hours.

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

# A   Theory for Transformer Construction

*Proof of Lemma 1.*  Recall from Section 2 the following expression for smax:

$$[\texttt{smax}\left(Z^\top W_\ell^{K^\top} W_\ell^Q Z\right)]_{ij} \propto \exp\left((W_\ell^K z^{(i)})^\top (W_\ell^Q z^{(j)})\right)$$

$$[\texttt{smax}(\texttt{logrbf}(W_\ell^K Z, W_\ell^Q Z))]_{ij} \propto \exp\left((W_\ell^K z^{(i)})^\top (W_\ell^Q z^{(j)}) - \|W_\ell^K z^{(i)}\|_2^2/2 - \|W_\ell^Q z^{(j)}\|_2^2/2\right).$$

In particular, the $\propto$ symbol is applied for fixed column $j$, across rows $i$.

Under our assumptions that $W_\ell^K = a I_{d \times d}$ and $W_\ell^Q = b I_{d \times d}$, we can simplify

$$(W_\ell^K z^{(i)})^\top (W_\ell^Q z^{(j)}) = ab z^{(i)^\top} z^{(j)}$$
$$\|W_\ell^K z^{(i)}\|_2^2/2 = a^2 \|z^{(i)}\|_2^2/2 = a^2 C^2$$
$$\|W_\ell^Q z^{(j)}\|_2^2/2 = b^2 \|z^{(j)}\|_2^2/2 = b^2 C^2$$

Carefully writing the expression for the smax term in $\texttt{Attn}_\texttt{rbf}$ gives

$$[\texttt{smax}(\texttt{logrbf}(W_\ell^K Z, W_\ell^Q Z))]_{ij}$$

$$= \frac{\exp\left((W_\ell^K z^{(i)})^\top (W_\ell^Q z^{(j)}) - \|W_\ell^K z^{(i)}\|_2^2/2 - \|W_\ell^Q z^{(j)}\|_2^2/2\right)}{\sum_{k \neq j}(W_\ell^K z^{(k)})^\top (W_\ell^Q z^{(j)}) - \|W_\ell^K z^{(k)}\|_2^2/2 - \|W_\ell^Q z^{(j)}\|_2^2/2}$$

$$= \frac{\exp\left(ab z^{(i)^\top} z^{(j)} - (a^2 + b^2)C^2\right)}{\sum_{k \neq j} \exp\left(ab z^{(k)^\top} z^{(j)} - (a^2 + b^2)C^2\right)}$$

$$= \frac{\exp\left(ab z^{(i)^\top} z^{(j)}\right)}{\sum_{k \neq j} \exp\left(ab z^{(k)^\top} z^{(j)}\right)}$$

$$= [\texttt{smax}(Z^\top W_\ell^{K^\top} W_\ell^Q Z)]_{ij}.$$

This concludes our proof.  □

*Proof of Lemma 2.*  Recall (6), where $(M_{\mathcal{F}})_{jj} = \mathbb{1}\{j \notin \mathcal{F}\}$. If $j \in \mathcal{F}$, then $(M_{\mathcal{F}})_{jj} = 0$ and the update gives $Z_{\ell+1}^{(j)} = Z_\ell^{(j)}$, matching the frozen-token rule in (4).

Now fix $j \notin \mathcal{F}$ and let $z_\ell^{(j)}$ denote the $j^{th}$ column of $Z_\ell$. Since $(M_{\mathcal{F}})_{jj} = 1$, (6) gives

$$z_{\ell+1}^{(j)} = \left[W_\ell^S Z_\ell\right]^{(j)} + \left[\texttt{Attn}_\texttt{rbf}\left(Z_\ell; W_\ell^V, W_\ell^Q, W_\ell^K\right)\right]^{(j)}.$$

With $W_\ell^Q = W_\ell^K = \frac{1}{\sigma_\ell} I_{d \times d}$, the RBF similarity between key $i$ and query $j$ is $\exp(-\|z_\ell^{(i)} - z_\ell^{(j)}\|_2^2/(2\sigma_\ell^2))$, and the column-wise softmax therefore equals $\kappa_\ell(i, j)$ in (5). Thus, using $W_\ell^V = \gamma_\ell I_{d \times d}$,

$$\left[\texttt{Attn}_\texttt{rbf}\left(Z_\ell; W_\ell^V, W_\ell^Q, W_\ell^K\right)\right]^{(j)} = \gamma_\ell \sum_{i \neq j} \kappa_\ell(i, j) z_\ell^{(i)}.$$

Using $W_\ell^S = (1 - \alpha_\ell - \gamma_\ell) I_{d \times d}$, we obtain

$$z_{\ell+1}^{(j)} = (1 - \alpha_\ell - \gamma_\ell) z_\ell^{(j)} + \gamma_\ell \sum_{i \neq j} \kappa_\ell(i, j) z_\ell^{(i)}$$

$$= (1 - \alpha_\ell) z_\ell^{(j)} + \gamma_\ell \sum_{i \neq j} \kappa_\ell(i, j)\left(z_\ell^{(i)} - z_\ell^{(j)}\right),$$

which is exactly (4).  □

## B  Theory for Manifold Denoising

*Proof of Lemma 3.*  Recall $\mathcal{L} = I_{n\times n} - \mathcal{D}^{-1}\mathcal{W}$, where $\mathcal{W}_{ij} := \exp(-\|z^{(i)} - z^{(j)}\|_2^2/(2\sigma^2))$ and $\mathcal{W}_{ii} = 0$, and $\mathcal{D}_{ii} = \sum_{j=1}^n \mathcal{W}_{ij}$. Then for each $i$,

$$\sum_{j=1}^n \mathcal{L}_{ij}\, z_\ell^{(j)} = z_\ell^{(i)} - \frac{1}{\mathcal{D}_{ii}}\sum_{j\neq i}\mathcal{W}_{ij}z_\ell^{(j)}.$$

Plugging this into (8) gives

$$z_{\ell+1}^{(i)} = z_\ell^{(i)} - \delta_\ell\Big(z_\ell^{(i)} - \frac{1}{\mathcal{D}_{ii}}\sum_{j\neq i}\mathcal{W}_{ij}z_\ell^{(j)}\Big) = (1-\delta_\ell)z_\ell^{(i)} + \delta_\ell\sum_{j\neq i}\frac{\mathcal{W}_{ij}}{\mathcal{D}_{ii}}z_\ell^{(j)}.$$

Since $\sum_{j\neq i}\mathcal{W}_{ij}/\mathcal{D}_{ii} = 1$, this can be rewritten as

$$z_{\ell+1}^{(i)} = z_\ell^{(i)} + \delta_\ell\sum_{j\neq i}\frac{\mathcal{W}_{ij}}{\mathcal{D}_{ii}}\Big(z_\ell^{(j)} - z_\ell^{(i)}\Big).$$

Noting that $\mathcal{W}_{ij}/\mathcal{D}_{ii} = \kappa_\ell(j,i)$ with $\sigma_\ell = \sigma$, we obtain (4) with $\mathcal{F} = \varnothing$, $\alpha_\ell = 0$, and $\gamma_\ell = \delta_\ell$. $\quad\square$

## C  Theory for Score-Based Diffusion (Section 4)

*Proof of Lemma 4.*  Let $x^{(1)},\dots,x^{(n)}$ denote the training set of unperturbed samples. Let $x^{(n+1)}$ denote the query sample. Let $z_0^{(i)} = x^{(i)}$ for $i = 1,\dots,n$. Let $z_0^{(n+1)} = x^{(n+1)} + \mathcal{N}(0,\sigma_0^2 I)$, where $\sigma_0^2 := T$ is the initial noise level. From (11),

$$z_{\ell+1}^{(i)} = z_\ell^{(i)} \qquad\qquad\qquad\qquad\qquad \text{for } i = 1,\dots,n$$

$$z_{\ell+1}^{(n+1)} = z_\ell^{(n+1)} + \frac{\delta_\ell}{2}\nabla\log p_{\sigma_\ell^2}(z_\ell^{(n+1)})$$

$$= z_\ell^{(n+1)} + \frac{\delta_\ell}{2\sigma_\ell^2}\frac{\sum_{i=1}^n \exp\Big(-\frac{1}{2\sigma_\ell^2}\big\|z_\ell^{(n+1)} - x^{(i)}\big\|_2^2\Big)\Big(x^{(i)} - z_\ell^{(n+1)}\Big)}{\sum_{i=1}^n \exp\Big(-\frac{1}{2\sigma_\ell^2}\big\|z_\ell^{(n+1)} - x^{(i)}\big\|_2^2\Big)}$$

$$= \Big(1 - \frac{\delta_\ell}{2\sigma_\ell^2}\Big)z_\ell^{(n+1)} + \frac{\delta_\ell}{2\sigma_\ell^2}\frac{\sum_{i=1}^n \exp\Big(-\frac{1}{2\sigma_\ell^2}\big\|z_\ell^{(n+1)} - x^{(i)}\big\|_2^2\Big)x^{(i)}}{\sum_{i=1}^n \exp\Big(-\frac{1}{2\sigma_\ell^2}\big\|z_\ell^{(n+1)} - x^{(i)}\big\|_2^2\Big)} \qquad (17)$$

Where the second equality is by (11), and the third equality is by the expression of the score in (12).

Comparing (17) to (4), we see that it is equivalent to letting $\mathcal{F} = \{z^{(1)},\dots,z^{(n)}\}$, and $\sigma_\ell = \sigma_\ell$, and $\gamma_\ell = \frac{\delta_\ell}{2\sigma_\ell^2}$. Consequently, let

$$Z_0 := [z^{(n+1)}] \in \mathbb{R}^{d\times 1}, \qquad Z^{\mathcal{F}} := \Big[z^{(1)},\dots,z^{(n)}\Big] \in \mathbb{R}^{d\times n}.$$

From (7),

$$Z_{\ell+1} = W_\ell^S Z_\ell + \texttt{CrAttn}_{\texttt{rbf}}\Big(Z^{\mathcal{F}}, Z_\ell; W_\ell^{V'}, W_\ell^{Q'}, W_\ell^{K'}\Big),$$

where we leave out the $\texttt{Attn}_{\texttt{rbf}}$ term because $Z_\ell$ contains the single token query $z_\ell^{(n+1)}$, and we assumed in Section 2 that $\texttt{Attn}$ does not have tokens attending to themselves.

By the lemma statement, we choose

$$W_\ell^V = \gamma_\ell = \frac{\delta_\ell}{2\sigma_\ell^2}I_{d\times d}, \qquad W_\ell^Q = W_\ell^K = \frac{1}{\sigma_\ell}I_{d\times d}, \qquad W_\ell^S = 1 - \frac{\delta_\ell}{2\sigma_\ell^2}.$$

Recall from Section 2.1 that $\texttt{CrAttn}_{\texttt{rbf}}(U, Z; W_\ell) := W_\ell^V U \, \texttt{smax}(\texttt{logrbf}(W_\ell^K U, W_\ell^Q Z))$. Here, $U = Z^{\mathcal{F}} = \{[z^{(1)}, \ldots, z^{(n)}]\}$ and $Z = Z_\ell$. Thus

$$
\begin{aligned}
Z_{\ell+1} &= \left(1 - \frac{\delta}{2\sigma_\ell^2}\right) Z_\ell + \texttt{CrAttn}_{\texttt{rbf}}\left(Z^{\mathcal{F}}, Z_\ell; W_\ell^{V'}, W_\ell^{Q'}, W_\ell^{K'}\right) \\
&= \left(1 - \frac{\delta}{2\sigma_\ell^2}\right) Z_\ell + \frac{\delta_\ell}{2\sigma_\ell^2} Z^{\mathcal{F}} \texttt{smax}\left(\texttt{logrbf}\left(\frac{1}{\sigma_\ell} Z^{\mathcal{F}}, \frac{1}{\sigma_\ell} Z_\ell\right)\right).
\end{aligned}
$$

Under the definition of $\texttt{CrAttn}$, $Z^{\mathcal{F}}$, and $Z_\ell = [z_\ell^{(n+1)}]$, we verify that the above is identical to (17).

$\square$

**Lemma 6** (Continuity Equation). *Let $p_t = p_0 * \mathcal{N}(0, tI)$. Its continuity equation is satisfied by $v_t = -\frac{1}{2}\nabla \log p_t(x)$, as defined in (10).*

*Proof of Lemma 6.* Because the forward SDE is standard Brownian motion, its density satisfies the heat equation[2] $\partial_t p_t = \frac{1}{2}\Delta p_t$. Define the velocity field $v_t(x) := -\frac{1}{2}\nabla \log p_t(x)$. Then, using the identity $p_t \nabla \log p_t = \nabla p_t$,

$$
-\nabla \cdot (p_t v_t) = -\nabla \cdot \left(-\frac{1}{2} p_t \nabla \log p_t\right) = \frac{1}{2}\nabla \cdot (\nabla p_t) = \frac{1}{2}\Delta p_t = \partial_t p_t,
$$

which is exactly (10).

$\square$

**Lemma 7.** *Consider the setting in Section 4.3. Assume that $\|z^{(i)}\|_2 \leq R$ for all $i$. Then for any $\|z\| \leq R$, we have*

$$
\mathbb{E}\left[\left\|\sum_{j=1}^{\tau} \frac{e^{-\|\psi^{(j)} - z\|_2^2/(2\sigma^2)}}{\sum_{k=1}^{\tau} e^{-\|\psi^{(k)} - z\|_2^2/(2\sigma^2)}}\left(\psi^{(j)} - z\right) - \sum_{j=1}^{n} \frac{e^{-\|z^{(j)} - z\|_2^2/(2\sigma^2)}}{\sum_{k=1}^{n} e^{-\|z^{(k)} - z\|_2^2/(2\sigma^2)}}\left(z^{(j)} - z\right)\right\|_2^2\right] = O(\tau^{-1})
$$

*Proof.* For convenience,

$$
① = \sum_{j=1}^{\tau} \frac{e^{-\|\psi^{(j)} - z\|_2^2/(2\sigma^2)}}{\sum_{k=1}^{\tau} e^{-\|\psi^{(k)} - z\|_2^2/(2\sigma^2)}}\left(\psi^{(j)} - z\right)
$$

$$
② = \sum_{j=1}^{n} \frac{e^{-\|z^{(j)} - z\|_2^2/(2\sigma^2)}}{\sum_{k=1}^{n} e^{-\|z^{(k)} - z\|_2^2/(2\sigma^2)}}\left(z^{(j)} - z\right)
$$

Let $A_\tau := \frac{1}{\tau}\sum_{j=1}^{\tau} e^{-\|\psi^{(j)} - z\|_2^2/(2\sigma^2)}\left(\psi^{(j)} - z\right)$ and $B_\tau = \frac{1}{\tau}\sum_{j=1}^{\tau} e^{-\|\psi^{(j)} - z\|_2^2/(2\sigma^2)}$. Analogously let $A_n := \frac{1}{n}\sum_{j=1}^{n} e^{-\|z^{(j)} - z\|_2^2/(2\sigma^2)}\left(z^{(j)} - z\right)$ and $B_n := \frac{1}{n}\sum_{k=1}^{n} e^{-\|z^{(k)} - z\|_2^2/(2\sigma^2)}$. Under our assumptions, there exists constants $C, a, a'$ such that $e^{-\|z^{(i)} - z\|_2^2/(2\sigma^2)}\left(z^{(j)} - z\right) \leq C$, and $a \leq e^{-\|z^{(i)} - z\|_2^2/(2\sigma^2)} \leq a'$

We can decompose

$$
\frac{A_\tau}{B_\tau} - \frac{A_n}{B_n} = \frac{A_\tau - A_n}{B_n} + A_\tau\left(\frac{1}{B_\tau} - \frac{1}{B_n}\right). \tag{18}
$$

The first term can be bound as

$$
\frac{\mathbb{E}\left[\|A_\tau - A_n\|_2^2\right]}{B_n} \leq \frac{1}{\tau B_n}\mathbb{E}\left[\|A_1 - A_n\|_2^2\right] \leq \frac{C^2}{\tau B_n} = O(\tau^{-1})
$$

---

[2]For unit-variance Brownian motion the infinitesimal generator is $\frac{1}{2}\Delta$, hence $\partial_t p_t = \frac{1}{2}\Delta p_t$.

The second term can be bound using a Taylor expansion of $f(r) = 1/r$:

$$f(r+c) = \frac{1}{r} - \frac{c}{r^2} + \int_r^{r+c} \int_r^s \frac{1}{2s^3} dt ds$$

$$\Rightarrow \quad \left| f(r+c) - f(r) + \frac{c}{r^2} \right| \le \frac{c^2}{\min\{r, r+c\}^3}$$

Thus we can bound the second term of (18) as

$$\mathbb{E}\left[ \left\| A_\tau \left( \frac{1}{B_\tau} - \frac{1}{B_n} \right) \right\|_2^2 \right] \le C^2 \mathbb{E}\left[ \left| \frac{1}{B_\tau} - \frac{1}{B_n} \right|_2^2 \right]$$

$$\le \frac{C^2}{B_n} \mathbb{E}\left[ (B_\tau - B_n)^2 \right] + \frac{C^2}{\min\{B_n, B_\tau\}^6} \mathbb{E}\left[ (B_\tau - B_n)^4 \right]$$

$$\le \frac{C^2 \mathbb{E}\left[ (B_1 - B_n)^2 \right]}{a\tau} + \frac{C^2 \mathbb{E}\left[ (B_1 - B_n)^4 \right]}{\tau^2 a^6}$$

$$\le \frac{C^2 a'^2}{a\tau} + \frac{C^2 a'^4}{\tau^2 a^6} = O(\tau^{-1})$$

$\square$

## C.1 Anisotropic Diffusion

*Proof of Lemma 5.* Let $x_0 \sim p_0$, and $dx_t = A_t dB_t$ for some matrix-valued function $t \to A_t$. For simplicity, assume that $A_t$ is symmetric, so that $A_t A_t^\top = A_t^2$. By the Fokker Planck equation,

$$\frac{\partial}{\partial t} p_t(x) = \frac{1}{2} \operatorname{tr}\left( A_t^2 \nabla^2 p_t \right) = \frac{1}{2} \mathbf{div}\left( p_t(x) A_t^2 \nabla \log p_t \right)$$

From the RHS, we verify that the continuity equation is satisfied by $dx_t = -\frac{1}{2} A_t^2 \nabla \log p_t(x)$. The forward ODE is thus given by

$$dx_t = -\frac{1}{2} A_t^2 \nabla \log p_t(x).$$

Let $p_t$ denote the distribution of $x_t$. Notice that the total Gaussian covariance under $dx_t = A_t dB_t$ is $M_t := \int_0^t A_s^2 ds$. Thus $p_t = p_0 * \mathcal{N}(0, M_t)$ The explicit form of $\nabla \log p_t(x)$ is

$$\nabla \log p_t(x) = M_t^{-1} \frac{\int \exp\left( -(x - x_0)^\top (2M_t)^{-1} (x - x_0) \right) (x_0 - x) dp_0(x_0)}{\int \exp\left( -(x - x_0)^\top (2M_t)^{-1} (x - x_0) \right) dp_0(x_0)}.$$

Therefore, the forward ODE is also more explicitly written as

$$dx_t = -\frac{1}{2} A_t^2 M_t^{-1} \underbrace{\frac{\int \exp\left( -(x - x_0)^\top (2M_t)^{-1} (x - x_0) \right) (x_0 - x) dp_0(x_0)}{\int \exp\left( -(x - x_0)^\top (2M_t)^{-1} (x - x_0) \right) dp_0(x_0)}}_{F(M_t, x)}$$

$$dM_t = A_t^2 dt.$$

Let $y_t = x_{T-t}$ denote the time-reversal of the forward ODE. Then the Euler-discretization of $y_t$ is given by

$$t_{\ell+1} = t_\ell - \delta_\ell$$

$$y_{\ell+1} = y_\ell + \frac{\delta_\ell}{2} A_{t_\ell}^2 M_{t_\ell}^{-1} F(M_{t_\ell}, x) \tag{19}$$

$$M_{t_{\ell+1}} = M_{t_\ell} - \delta_\ell A_{t_\ell}^2$$

In the above, $A_{t_\ell}$ is a sequence that we assume is known a priori. $M_{t_\ell}$ can be computed from $A_{t_\ell}$ via the last line. $M_0$ denotes the initial Gaussian noise covariance, thus $M_0 = TI$. Subsequently, we simplify the notation $M_{t_\ell} \to M_\ell$ and $A_{t_\ell} \to A_\ell$.

In the score-based denoising setup, we have $p_0$ be a discrete distribution over the training set $x^{(1)}, \ldots, x^{(n)}$. Thus

$$F(M, x) = \frac{\sum_{i=1}^{n} \exp\left(-(x - x^{(i)})^{\top}(2M)^{-1}(x - x^{(i)})\right)(x^{(i)} - x)}{\sum_{i=1}^{n} \exp\left(-(x - x^{(i)})^{\top}(2M)^{-1}(x - x^{(i)})\right)}$$

Let $Z^{\mathcal{F}} := \left[z^{(1)}, \ldots, z^{(n)}\right]$, and $Z_\ell := [z_\ell^{(n+1)}]$. Observe that we are trying to denoise the query $z_\ell^{(n+1)}$, so $z_\ell^{(n+1)}$ is $y_\ell$ in (19). Therefore, (19) is equivalent to

$$
\begin{aligned}
z_{\ell+1}^{(n+1)} =& z_\ell^{(n+1)} + \frac{\delta_\ell A_\ell^2 M_\ell^{-1}}{2} F(M_\ell, z_\ell^{(n+1)}) \\
=& z_\ell^{(n+1)} + \frac{\delta_\ell A_\ell^2 M_\ell^{-1}}{2} Z^{\mathcal{F}} \, \mathtt{smax}\left(\mathtt{logrbf}\left(M_\ell^{-1/2} Z^{\mathcal{F}}, M_\ell^{-1/2} Z_\ell\right)\right) - \frac{\delta_\ell A_\ell^2 M_\ell^{-1} z_\ell^{(n+1)}}{2},
\end{aligned}
$$

where the last line can be verified by definition of $\mathtt{smax}$ and $\mathtt{rbf}$. Observe that $z_\ell^{(n+1)}$ above is equal to $Z_\ell$.

By pattern matching, we see that the above coincides with the choices of

$$W_\ell^V = \frac{\delta_\ell A_\ell^2 M_\ell^{-1}}{2}, \qquad W_\ell^Q = W_\ell^K = M_\ell^{-1/2}, \qquad W_\ell^S = \left(I - \frac{\delta_\ell A_\ell^2 M_\ell^{-1}}{2}\right),$$

which is exactly the parameter setting in the paper. $\qquad\square$

## D  Miscellaneous Theory

The following standard result guarantees that Gaussian distributions, and the sum between a Gaussian distribution and an arbitrary unit-length vector, lie on a sphere.

**Lemma 8** (Radius concentration under Gaussian perturbation.). *Let $v \in \mathbb{S}^{d-1}$. Let $u \sim \mathcal{N}(0, cI_{d \times d})$. Then for $t \in (0, 1)$,*

$$\mathbb{P}\left(\left|\|u\|_2^2 - cd\right| \geq tcd\right) \leq 2e^{-dt^2/8}$$

$$\mathbb{P}\left(\left|\|v + u\|_2^2 - (1 + c(d-1))\right| \geq tcd\right) \leq 2e^{-dt^2/16}$$

*Proof of Lemma 8.* The first equality is simply follows by $(2\sqrt{d}, 4)$-sub-exponential concentration of $d$-dimensional Gaussians.

The second equality follows by decomposing $u$ into $u_\parallel = \left\langle u, \frac{v}{\|v\|_2} \right\rangle \frac{v}{\|v\|_2}$ and $u_\perp = u - u_\parallel$ By the same sub-exponential bound applied to $u_\perp$, we have

$$\mathbb{P}\left(\left|\|u_\perp\|_2^2 - c(d-1)\right| \geq tcd/2\right) \leq 2e^{-dt^2/16}$$

On the other hand, we have

$$\mathbb{P}\left(\left|\|v + u_\parallel\|_2^2 - 1\right| \geq tcd/2\right) = \mathbb{P}\left(\|u_\parallel\|_2^2 \geq tcd/2\right) \leq 2e^{-d^2t^2/16}.$$

The second inequality then follows by union bound. $\qquad\square$

## E  Experiments

### E.1  General Experiment Details

In our experiments, we use both the MNIST (60,000 samples, 10 classes) and CIFAR10 (50,000, 10 classes) datasets. In each case, we use a train/test split of 9:1. All experiments are run on a single A5000 GPU.

In all our experiments, we do not use any positional encoding or tokenization of the image. Instead, we simply represent each image as a vector of all its pixel values. For MNIST, the vector dimension is $784 = 28 * 28$. For CIFAR10 and CIFAR100, the vector dimension is $3072 = 3 * 32 * 32$.

## E.2 Additional CIFAR-10/100 Metrics and FID

**FID protocol (minimal).** We report Fréchet Inception Distance (FID) using the *Inception-V3 pool_3* features (2048-D), with feature normalization enabled. Real images are the held-out *test split*; fake images are generated by the same pipeline used for the corresponding figure (e.g., per-layer denoising outputs when the figure plots layerwise losses). We compute FID per configuration and report the mean $\pm$ standard deviation over $k$ independent runs (same seeds used for CIFAR-10 and CIFAR-100 for parity). Unless otherwise noted, we use batch sizes matching the main experiments and evaluate on the full test split.

### E.2.1 Addendum to Fig. 3: score test loss — CIFAR-100 and FID

**What we add.** (i) CIFAR–100 analog of the loss curves (see Fig. 8); (ii) FID for CIFAR–10 and CIFAR–100 at each layer for the two trained variants (RBF, STD). We follow the same evaluation protocol as in App. E.2.

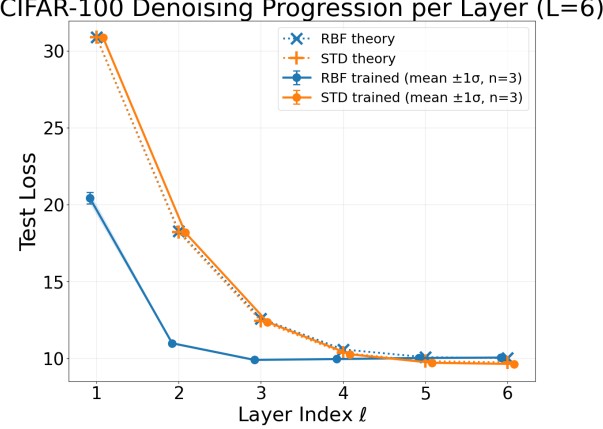

Figure 8: CIFAR–100 RMSE per layer for the curves in Fig. 3.

Table 1: Per-layer FID and RMSE for CIFAR–10 and CIFAR–100. Mean ± std over seeds.

| Dataset | Metric | Layer | RBF trained | STD trained |
|---|---|---|---|---|
| CIFAR10 | FID | 1 | $347.2712 \pm 2.4329$ | $403.9372 \pm 0.9406$ |
| | | 2 | $198.1683 \pm 1.4204$ | $321.1340 \pm 1.1317$ |
| | | 3 | $99.7453 \pm 1.6279$ | $233.5431 \pm 1.5499$ |
| | | 4 | $43.6949 \pm 1.2320$ | $163.7732 \pm 0.9192$ |
| | | 5 | $21.8445 \pm 0.4051$ | $114.8899 \pm 0.5935$ |
| | | 6 | $19.7110 \pm 0.0940$ | $78.7409 \pm 0.1956$ |
| | RMSE | 1 | $20.4389 \pm 0.3717$ | $30.8847 \pm 0.0038$ |
| | | 2 | $11.0475 \pm 0.0544$ | $18.2189 \pm 0.0016$ |
| | | 3 | $9.9866 \pm 0.0112$ | $12.4526 \pm 0.0037$ |
| | | 4 | $10.0475 \pm 0.0148$ | $10.3822 \pm 0.0064$ |
| | | 5 | $10.1168 \pm 0.0144$ | $9.8443 \pm 0.0080$ |
| | | 6 | $10.1454 \pm 0.0141$ | $9.7725 \pm 0.0087$ |
| CIFAR100 | FID | 1 | $327.7921 \pm 5.2556$ | $390.9332 \pm 1.5852$ |
| | | 2 | $181.1612 \pm 3.7589$ | $300.4900 \pm 2.2285$ |
| | | 3 | $95.6904 \pm 2.5033$ | $212.3552 \pm 2.2666$ |
| | | 4 | $47.0779 \pm 1.8721$ | $150.9074 \pm 1.3448$ |
| | | 5 | $25.1974 \pm 1.0298$ | $107.3123 \pm 1.1695$ |
| | | 6 | $22.5363 \pm 0.4453$ | $77.9261 \pm 0.9152$ |
| | RMSE | 1 | $20.4177 \pm 0.3744$ | $30.8679 \pm 0.0044$ |
| | | 2 | $10.9805 \pm 0.0717$ | $18.1712 \pm 0.0147$ |
| | | 3 | $9.8963 \pm 0.0532$ | $12.3639 \pm 0.0331$ |
| | | 4 | $9.9517 \pm 0.0570$ | $10.2620 \pm 0.0485$ |
| | | 5 | $10.0194 \pm 0.0577$ | $9.7092 \pm 0.0565$ |
| | | 6 | $10.0477 \pm 0.0578$ | $9.6317 \pm 0.0600$ |

**Takeaways** FID decreases monotonically with layer index on both datasets, indicating progressive denoising. Across layers, **RBF trained** attains consistently lower (better) FID than **STD trained**. For RMSE, both datasets converge near 10 at the final layer; **STD trained** ends slightly lower than **RBF trained** at layer 6 (CIFAR–10: 9.77 vs. 10.15; CIFAR–100: 9.63 vs. 10.05).

### E.3 Addendum to Fig. 5: Context vs. Loss on CIFAR-100 and FID on CIFAR-10/100

We report test-set FID and RMSE across context lengths. Values are mean ± one standard deviation over three seeds. Results mirror the trend in Fig. 5: modest context substantially lowers error, with diminishing returns at larger context.

Table 2: Addendum to Fig. 5: FID and RMSE vs. context length ($n_{ctx}$). Mean ± $1\sigma$ over 3 seeds.

| CIFAR-10 | | |
|---|---|---|
| $n_{ctx}$ | FID | RMSE |
| 0 | $368.79 \pm 22.66$ | $133.9387 \pm 6.7723$ |
| 2 | $102.12 \pm 130.07$ | $10.3142 \pm 0.7683$ |
| 6 | $14.06 \pm 0.71$ | $11.1293 \pm 0.0403$ |
| 16 | $30.14 \pm 14.65$ | $10.3618 \pm 0.4681$ |
| 32 | $31.80 \pm 11.76$ | $10.2821 \pm 0.4452$ |
| 60 | $47.28 \pm 0.81$ | $9.8351 \pm 0.0090$ |
| **CIFAR-100** | | |
| $n_{ctx}$ | FID | RMSE |
| 0 | $333.26 \pm 77.20$ | $150.0745 \pm 36.1599$ |
| 2 | $16.55 \pm 1.23$ | $11.0319 \pm 0.1125$ |
| 8 | $21.16 \pm 4.74$ | $10.7499 \pm 0.2683$ |
| 16 | $21.94 \pm 4.06$ | $10.6655 \pm 0.2589$ |
| 32 | $42.78 \pm 1.26$ | $9.9464 \pm 0.0324$ |
| 48 | $44.02 \pm 0.40$ | $9.9238 \pm 0.0206$ |

*Note.* FID decreases sharply with small context; RMSE shows smaller but consistent gains with additional context. CIFAR-100 exhibits the same qualitative pattern as CIFAR-10.

### E.3.1 Addendum to Fig. 7: witness/anisotropic — CIFAR-100 and FID

**What we add.** (i) CIFAR-100 loss vs. #witness tokens; (ii) FID and RMSE on CIFAR-10 and CIFAR-100 vs. $\tau$ for the *Witness+RBF* and *Witness+anisotropic* models. Baseline exact-score model metrics from Fig. 7 are also provided for comparison.

Table 3: Fig. 7 addendum: FID and RMSE at the final layer vs. #witness tokens ($\tau$) for **CIFAR-10**. Mean $\pm$ std over $k$ seeds.

| # Witnesses ($\tau$) | Witness+RBF | | Witness+Anisotropic | |
|---|---|---|---|---|
| | FID | RMSE | FID | RMSE |
| 20 | $339.00 \pm 0.70$ | $7.8919 \pm 0.0020$ | $306.09 \pm 0.34$ | $7.4240 \pm 0.0016$ |
| 40 | $360.82 \pm 0.13$ | $7.5876 \pm 0.0034$ | $240.09 \pm 0.60$ | $7.2523 \pm 0.0017$ |
| 100 | $312.56 \pm 0.02$ | $7.3838 \pm 0.0026$ | $193.31 \pm 0.56$ | $7.1351 \pm 0.0007$ |
| 200 | $267.91 \pm 1.01$ | $7.2784 \pm 0.0019$ | $173.66 \pm 0.24$ | $7.0598 \pm 0.0005$ |
| 400 | $219.15 \pm 0.86$ | $7.2307 \pm 0.0016$ | $161.80 \pm 0.70$ | $7.0427 \pm 0.0012$ |
| 800 | $239.34 \pm 0.18$ | $7.2414 \pm 0.0032$ | $168.74 \pm 0.64$ | $7.0427 \pm 0.0019$ |
| 1000 | $271.73 \pm 0.51$ | $7.2924 \pm 0.0014$ | $168.89 \pm 0.37$ | $7.0462 \pm 0.0003$ |
| **Baseline (Exact)** | 21.57 | 10.0805 | 23.57 | 9.5456 |

Table 4: Fig. 7 addendum: FID and RMSE at the final layer vs. #witness tokens ($\tau$) for **CIFAR-100**. Mean $\pm$ std over $k$ seeds.

| # Witnesses ($\tau$) | Witness+RBF | | Witness+Anisotropic | |
|---|---|---|---|---|
| | FID | RMSE | FID | RMSE |
| 20 | $313.06 \pm 0.16$ | $7.8954 \pm 0.0020$ | $284.66 \pm 6.37$ | $7.4390 \pm 0.0055$ |
| 40 | $337.84 \pm 0.84$ | $7.7204 \pm 0.0027$ | $221.24 \pm 0.46$ | $7.2714 \pm 0.0035$ |
| 100 | $306.16 \pm 0.14$ | $7.4170 \pm 0.0027$ | $181.34 \pm 0.36$ | $7.1272 \pm 0.0032$ |
| 200 | $262.20 \pm 0.22$ | $7.2780 \pm 0.0022$ | $168.53 \pm 0.06$ | $7.0650 \pm 0.0019$ |
| 400 | $222.98 \pm 0.98$ | $7.2226 \pm 0.0038$ | $164.91 \pm 0.22$ | $7.0323 \pm 0.0036$ |
| 800 | $236.45 \pm 0.14$ | $7.2424 \pm 0.0038$ | $163.78 \pm 0.34$ | $7.0244 \pm 0.0048$ |
| 1000 | $258.02 \pm 0.27$ | $7.2733 \pm 0.0034$ | $167.02 \pm 0.26$ | $7.0415 \pm 0.0036$ |
| **Baseline (Exact)** | 26.64 | 9.5048 | 26.53 | 9.5242 |

We discuss a few key observations from Tables 3 and 4:

**Exact Baseline: low FID, high RMSE.** RMSE measures per-image reconstruction, whereas FID measures distribution-level realism. The Baseline (Exact) model implements the exact score-based denoising via cross-attention to the full training set (Section 4.2); at test time, this dynamics pulls the noisy query toward an actual training example. This behavior is visible in Figure 4, where exact score denoising reproduces training images rather than the held-out target. Thus the model essentially "memorizes" the training set of real images, yielding excellent FID. However, this same behavior results in a worse RMSE – when the noisy test image's ground truth is not in the training set, the model selects a different point from the data manifold (i.e., a different training image), which is often far from the clean test image.

**Witness models: high FID, low RMSE.** In contrast, the **Witness+RBF** and **Witness+Anisotropic** models replace full-set cross-attention with a compact, learned witness set (Section 4.3 - E.4.2). This induces a geometry-aware score approximation that avoids instance-level copying, so the denoising trajectory tracks the actual test instance rather than a nearest training neighbor, i.e. better generalization, and thus lower test RMSE. The cost is higher FID: outputs are less tightly matched to the empirical sample distribution measured by Inception features. **Witness+Anisotropic** consistently outperforms **Witness+RBF**, reflecting the benefit of geometry-aware updates.

These results highlight a generalization–realism trade-off: the Exact Baseline optimizes distributional realism (FID) by memorizing the training set, whereas Witness models trade some FID for stronger generalization to unseen test inputs.

### E.4 Implementation Details for Specific Experiments

In this section, we describe details of several experiment setups and Transformer implementations:

1. The **in-context score-based denoising** experiment from Section 4.2 is explained in Appendix E.4.1.
2. The experiments comparing Witness-based denoising and Anisotropic Denoising from Section 4.5 is explained in Appendix E.4.2.

### E.4.1 Implementation Details for In-Context Score-based Denoising

In this section, we provide further details on the in-context score-based denoising experiment outlined in Section 4.2. For ease of reference, we repeat below the setup described in the main paper:

Input consists of the noiseless training samples $\{x^{(1)}, \ldots, x^{(n)}\}$, as well as $m$ **contextual images** $\{x^{(n+1)}, \ldots, x^{(n+m)}\}$. The query image is $x^{(n+m+1)}$. Importantly, all $m$ contextual images *belong to the same class as the query image* $x^{(n+m+1)}$. The Transformer input is $z^{(i)} = x^{(i)}$ for $i = 1, \ldots, n+m$, and the query point is perturbed with Gaussian noise: $z^{(n+m+1)} = x^{(n+m+1)} + \mathcal{N}(0, \sigma^2 I)$. We call $m$ the "context length". In terms of implementation, the query $x^{(n+m+1)}$ attends to the training set $x^{(1)}, \ldots, x^{(n)}$ via the `CrAttn` module in (7), while the context samples + query $x^{(n+1)}, \ldots, x^{(n+m)}$ attend to each other via the `Attn` module in (7) (with appropriate masking).

The Transformer architecture is as defined in (7):

$$Z_{\ell+1} = W_\ell^S Z_\ell + \texttt{Attn}_{\texttt{rbf}}\left(Z_\ell; W_\ell^V, W_\ell^Q, W_\ell^K\right) + \texttt{CrAttn}_{\texttt{rbf}}\left(Z^{\mathcal{F}}, Z_\ell; W_\ell^{V'}, W_\ell^{Q'}, W_\ell^{K'}\right).$$

We let

$$Z_0 = \left[x^{(n+1)}, \ldots, x^{(n+m)}, x^{(n+m+1)}\right] \in \mathbb{R}^{d \times (m+1)}$$

denote the context plus queries, and let

$$Z^{\mathcal{F}} = \left[x^{(1)}, \ldots, x^{(n)}\right] \in \mathbb{R}^{d \times n}$$

denote the frozen training set. Since the context tokens in the first $m$ columns of $Z_0$ are noiseless, we enforce that they are not updated using an additional mask. In summary,

$$Z_{\ell+1}^{(m+1)} = \left[W_\ell^S Z_\ell + \texttt{Attn}_{\texttt{rbf}}\left(Z_\ell; W_\ell^V, W_\ell^Q, W_\ell^K\right) + \texttt{CrAttn}_{\texttt{rbf}}\left(Z^{\mathcal{F}}, Z_\ell; W_\ell^{V'}, W_\ell^{Q'}, W_\ell^{K'}\right)\right]^{(m+1)}$$

$$Z_{\ell+1}^{(i)} = Z_\ell^{(i)} \qquad \text{for } i = 1, \ldots, m.$$

Note that $Z_{\ell+1}^{(i)}$ denotes the $i^{th}$ column of $Z_\ell$ and corresponds to token $z^{(n+i)}$.

Each parameter is constrained to be a scalar multiple of identity, i.e.

$$W_\ell^S = w_\ell^S I_{d \times d}, \quad W_\ell^Q = w_\ell^Q I_{d \times d}, \quad W_\ell^K = w_\ell^K I_{d \times d}, \quad W_\ell^V = w_\ell^V I_{d \times d}$$

$$W_\ell^{Q'} = w_\ell^{Q'} I_{d \times d}, \quad W_\ell^{K'} = w_\ell^{K'} I_{d \times d}, \quad W_\ell^{V'} = w_\ell^{V'} I_{d \times d},$$

so the total trainable parameters are $\left\{w_\ell^S, w_\ell^Q, w_\ell^K, w_\ell^V, w_\ell^{Q'}, w_\ell^{K'}, w_\ell^{V'}\right\}_{\ell=1,\ldots,L} \subset \mathbb{R}$.

### E.4.2 Implementation Details for Witness and Anisotropic+Witness Experiments

In this section, we provide further details on the in-context score-based denoising experiment outlined in Section 4.5. For ease of reference, we repeat below the setup described in the main paper:

Input consists of the noiseless training samples $\{x^{(1)}, \ldots, x^{(n)}\}$. The query image is $x^{(n+1)}$. The Transformer input is $z^{(i)} = x^{(i)}$ for $i = 1, \ldots, n$, and the query point is perturbed with Gaussian noise: $z^{(n+1)} = x^{(n+1)} + \mathcal{N}(0, \sigma^2 I)$. In terms of implementation, the query $x^{(n+1)}$ attends to the training set $x^{(1)}, \ldots, x^{(n)}$ via the `CrAttn` module in (14).

The Transformer architecture is as defined in (14):

$$Z_{\ell+1} = W_\ell^S Z_\ell + \texttt{CrAttn}_{\texttt{rbf}}\left(U_\ell, Z_\ell; W_\ell^{V'}, W_\ell^{Q'}, W_\ell^{K'}\right),$$

in the above, we leave out the $\texttt{Attn}$ module since it is not used. For all subsequent discussion, $U_\ell \in \mathbb{R}^{d \times S}$ is a trainable parameter, where $S$ is the number of witnesses in one layer. We initialize each $U_\ell$ by setting its $i^{th}$ column to be a randomly drawn training sample from $x^{(1)}, \ldots, x^{(n)}$. Note that each layer $\ell$ has a separate $U_\ell$.

For Witness+RBF, we have

$$Z_0 = \left[z^{(n+1)}\right] \in \mathbb{R}^{d \times 1},$$

$$Z_{\ell+1}^{(1)} = \left[W_\ell^S Z_\ell + \texttt{CrAttn}_{\texttt{rbf}}\left(U_\ell, Z_\ell; W_\ell^{V'}, W_\ell^{Q'}, W_\ell^{K'}\right)\right]^{(1)},$$

where $Z_{\ell+1}^{(1)}$ corresponds to the query token $z^{(n+1)}$. Each parameter is constrained to be a scalar multiple of identity, i.e.

$$W_\ell^S = w_\ell^S I_{d \times d}, W_\ell^{Q'} = w_\ell^{Q'} I_{d \times d}, \quad W_\ell^{K'} = w_\ell^{K'} I_{d \times d}, \quad W_\ell^{V'} = w_\ell^{V'} I_{d \times d},$$

so the total trainable parameters are the learned witnesses $\{U_\ell\}_{\ell=1,\ldots,L} \subset \mathbb{R}^{d \times S}$, and the (scalar) parameters $\left\{w_\ell^S, w_\ell^{Q'}, w_\ell^{K'}, w_\ell^{V'}\right\}_{\ell=1,\ldots,L} \subset \mathbb{R}$.

For Witness+Anisotropic, we have

$$Z_0 = \left[z^{(n+1)}\right] \in \mathbb{R}^{d \times 1},$$

$$Z_{\ell+1}^{(1)} = \left[W_\ell^S Z_\ell + \texttt{CrAttn}_{\texttt{rbf}}\left(U_\ell, Z_\ell; W_\ell^{V'}, W_\ell^{Q'}, W_\ell^{K'}\right)\right]^{(1)},$$

where $Z_{\ell+1}^{(1)}$ corresponds to the query token $z^{(n+1)}$. Each parameter is constrained to be a $d \times d$ diagonal matrix, so the total trainable parameters are the learned witnesses $\{U_\ell\}_{\ell=1,\ldots,L} \subset \mathbb{R}^{d \times S}$ and the diagonal parameter matrices $\left\{W_\ell^S, W_\ell^{Q'}, W_\ell^{K'}, W_\ell^{V'}\right\}_{\ell=1,\ldots,L} \subset \mathbb{R}^{d \times d}$ (each parameterized by the $d$ diagonal scalars).

## F  Future Directions

**Parameter-efficient denoising.**  Our characterization of anisotropic denoising with diagonal attention weights suggests that even highly constrained parameterizations can learn powerful denoising algorithms. This opens up opportunities for efficient Transformer variants in generative models— for example, using low-rank or diagonal attention in diffusion Transformers, or fine-tuning these components in parameter-efficient learning settings.

**Localized, interpretable attention.**  We observe that trained Transformers often specialize different layers (or heads) to denoise approximately non-overlapping semantic patches (Figure 1). This suggests a promising direction for designing sparse, locality-aware attention mechanisms in structured domains like vision. Such mechanisms may be more interpretable and robust to noisy or incomplete context, and they align with the geometry-aware updates formalized in our analysis.

**Multi-modal Contexts**  Incorporating conditioning signals (e.g., text) via cross-attention into our kernel-weighted update view provides a principled path to conditional denoising and controllable generation, and can be combined with parameter-efficient anisotropic modules.

## G  Limitations:

Our analysis and experiments deliberately focus on small-scale image datasets and small-depth, single-head Transformers. While this choice isolates the denoising mechanisms we study, it leaves open how the same constructions behave on larger, more varied data (e.g., high-resolution images, audio, or text) and in deeper, multi-stage architectures. Second, we evaluate only pixel-space RMSE; assessing perceptual quality, class fidelity, or downstream generation tasks remains future work.

