# OpenReview forum: "From Softmax to Score: Transformers Can Effectively Implement In-Context Denoising Steps"
_NeurIPS.cc/2025/Conference — NeurIPS 2025 poster_

### Official Review · Reviewer_Xi3K · 2025-06-30

**Clarity:** 3
**Significance:** 2
**Originality:** 3
**Rating:** 4
**Confidence:** 2

**Summary:**

The authors of this paper present a theoretical interpretation of transformers from the perspective of denoising processes in diffusion-based generation. They propose a specifically defined forward diffusion process that enables the reverse process to be formulated in a manner equivalent to the kernel-weighted update scheme of transformers, which can be implemented through the attention mechanism inherent to transformer architectures.

Building on this theoretical insight, the authors conduct experiments to empirically support their findings. Their approach is validated across various settings, including single-image generation, in-context class-conditioned generation, scenarios with a limited set of learnable witnesses, and anisotropic diffusion settings.

**Questions:**

1. In Figure 7(a), it appears counterintuitive that the Witness + RBF configuration outperforms the baseline as the number of witnesses increases. From my understanding, the baseline represents a model without the constraints of a limited witness set and, thus, should theoretically serve as the upper bound for the performance achievable by the witness-based model. Could the authors clarify how the Witness + RBF setting surpasses the baseline in this case? Is there an underlying factor, such as regularization effects or inductive bias introduced by the witness set, that explains this observation?

**Ethical Concerns:**

["NO or VERY MINOR ethics concerns only"]

**Final Justification:**

Initially, I had some concerns, but they were mostly resolved through the authors' rebuttal. After reviewing other reviewers' comments and participating in the discussion, I came to better appreciate the value of the paper and have decided to raise my score accordingly.

**Limitations:**

yes

**Paper Formatting Concerns:**

no formatting concerns

**Quality:**

3

**Strengths And Weaknesses:**

### **Strength**

**1. Novel interpretation of transformer computation as a denoiser**

The authors provide a theoretical analysis demonstrating that transformer computations can inherently act as a denoiser within the diffusion generation framework. This theoretical claim is further supported by empirical results on MNIST and CIFAR-10 generation tasks, confirming its practical feasibility.


### **Weaknesses**

**1. Insufficient alignment between theory and practical implementation**

The discretized sampling process described in Equation (10) requires a significant number of iterative steps to properly implement the denoising process. However, the practical transformer-based implementation utilizes only 6 layers, which may be insufficient to faithfully approximate the theoretical framework even compared with previous diffusion based methods. This discrepancy raises concerns about the strict validity of the theoretical-to-practical alignment.

**2. Non-differentiability of the witness set selection process**

It remains unclear how the selection of elements from the set $U_{I}$ (the small set of learnable witnesses) is performed in a differentiable manner. If the witness set selection occurs within the training process, a clear explanation of how this selection can be integrated with gradient-based optimization is required to ensure the method's consistency and practical applicability.

**3. Limited evaluation of generation performance**

While test loss provides a useful proxy for evaluating model performance on the test set, it does not fully reflect the effectiveness of the model as a diffusion denoiser. Standard generation quality metrics, such as Fréchet Inception Distance (FID) or Inception Score (IS), should be reported to quantitatively assess the quality and diversity of the generated images. The absence of these metrics weakens the empirical validation of the proposed method.

---

> ### Author Rebuttal · Authors · 2025-07-31
>
> We thank the reviewer for their thoughtful comments and suggestions. Below, we address the main points of concern, organized thematically for clarity.
>
> ## 1. Insufficient alignment between theory and practical implementation
> > The discretized sampling process described in Equation (10) requires a significant number of iterative steps to properly implement the denoising process. However, the practical transformer-based implementation utilizes only 6 layers, which may be insufficient to faithfully approximate the theoretical framework even compared with previous diffusion based methods. This discrepancy raises concerns about the strict validity of the theoretical-to-practical alignment.
>
> We thank the reviewer for this thoughtful observation. We agree that, in general, fewer denoising steps may increase discretization error in reverse-time diffusion processes.
>
>
> - We note that in Equation (10), **the discretization step size can be made larger to reduce the required number of steps**. In our implementation, we choose a step size that enables the process to be approximated within 6 layers. As shown in Figures 2 and 3, the test loss decreases monotonically with depth, indicating that even with this coarse discretization, the Transformer continues to refine its denoising behavior.
> - Our primary goal is to **understand the mechanistic capability of attention layers to implement denoising dynamics**—not necessarily to match state-of-the-art generative quality. From this perspective, using a small number of layers allows us to clearly attribute performance improvements to specific architectural choices.
> - **Our theoretical result in Lemma 4 holds for arbitrarily deep Transformers**, and our design uses the same parameter configuration across all layers. This naturally opens the door to recurrent implementations (e.g., looped Transformers), where the same layer is reused multiple times to achieve arbitrarily fine discretization. Exploring this direction is an exciting avenue for future work.
>
>
>
> ## 2. Non-differentiability of the witness set selection process
> > It remains unclear how the selection of elements from the set $U_I$ (the small set of learnable witnesses) is performed in a differentiable manner. If the witness set selection occurs within the training process, a clear explanation of how this selection can be integrated with gradient-based optimization is required to ensure the method's consistency and practical applicability.
>
> Thank you for bringing up this point. Prior to training, we sample a random subset $\psi^{(1)}...\psi^{(\tau)}$ of size $\tau$ from the training set (of size $n$), and use $\psi^{(1)}...\psi^{(\tau)}$ to initialize $U_\ell \in \mathbb{R}^{d\times \tau}$. **Subsequently, we treat the *entire $U_\ell \in \mathbb{R}^{d\times \tau}$* as a learnable parameter in continuous space, and optimize it using gradient-descent**. At convergence of training, the columns of $U_\ell$ do not need to be elements from the training set (e.g. each column of $U_\ell$ could be an average over multiple training samples.)
>
> The above process is defined on lines 234-241, and we will add further clarifications in the next draft. We also remark that the optimal choice of $U_\ell$ has connections to several kernel-related algorithms such as the Nystrom method, as detailed in [5,9,14,21] referenced in the paper.
>
> ## 3. Limited evaluation of generation performance
> > While test loss provides a useful proxy for evaluating model performance on the test set, it does not fully reflect the effectiveness of the model as a diffusion denoiser. Standard generation quality metrics, such as Fréchet Inception Distance (FID) or Inception Score (IS), should be reported to quantitatively assess the quality and diversity of the generated images. The absence of these metrics weakens the empirical validation of the proposed method.
>
> We thank the reviewer for this valuable suggestion. In response, we have computed the Fréchet Inception Distance (FID) across varying Transformer depths. The results for CIFAR-10 and CIFAR-100 are reported in the table below. As expected, the FID improves substantially with network depth and is consistent across datasets.
>
> We note that the absolute FID values are significantly higher than those of state-of-the-art generative models. This discrepancy is expected due to the minimalistic nature of our architecture:
>
> 1. **No positional encoding**: our model does not possess spatial inductive bias and cannot exploit pixel locality.
> 2. **No image tokenization**: images are input as flattened vectors, so attention must operate over raw pixel vectors.
> 3. **No multi-head attention, MLP, or full-rank projection layers**: the attention matrices $W^Q, W^K, W^V$ are diagonal, and the Transformer uses only a single head and 6 layers.
>
> By contrast, state-of-the-art models employ deep, multi-head Transformers or UNet-style CNNs with convolutional inductive bias, positional encodings, and patch-based input representations that are specifically designed to optimize perceptual metrics such as FID.
>
> Our focus in this paper is on **mechanistically understanding the denoising capabilities of the attention mechanism in isolation**. This motivates our use of RMSE as a primary metric: it directly evaluates denoising fidelity without entangling architectural heuristics. Nevertheless, we agree that FID is a useful complementary signal, and we view incorporating more perceptual metrics into future, more expressive variants of our architecture as an exciting future direction.
>
> ### CIFAR10/CIFAR100 FID-vs-Layer
> | Layer | CIFAR-10 (Theory) | CIFAR-10 (Trained) | CIFAR-100 (Theory) | CIFAR-100 (Trained) |
> | ----- | ----------------- | ------------------ | ------------------ | ------------------- |
> | 1     | 405               | 343.97 ± 1.36      | 392                | 324.51 ± 2.15       |
> | 2     | 325               | 195.97 ± 1.86      | 301                | 179.15 ± 1.44       |
> | 3     | 238               | 98.03 ± 1.00       | 217                | 95.07 ± 0.91        |
> | 4     | 167               | 43.26 ± 0.77       | 155                | 46.24 ± 1.06        |
> | 5     | 113               | 21.87 ± 0.54       | 108                | 25.25 ± 0.69        |
> | 6     | 78.3              | 19.31 ± 0.15       | 76.6               | 22.88 ± 0.38        |
>
>
> ## 4. Question: why does witness+RBF surpass baseline?
> > In Figure 7(a), it appears counterintuitive that the Witness + RBF configuration outperforms the baseline as the number of witnesses increases. From my understanding, the baseline represents a model without the constraints of a limited witness set and, thus, should theoretically serve as the upper bound for the performance achievable by the witness-based model. Could the authors clarify how the Witness + RBF setting surpasses the baseline in this case? Is there an underlying factor, such as regularization effects or inductive bias introduced by the witness set, that explains this observation?
>
> This is a sharp observation. This is related to your question above about how $U_\ell$ is optimized. We conjecture that the witness-based model achieves better performance than the baseline because **it has the freedom to select (and attend to) witness-vectors outside the training set,** whereas the baseline can only attend to samples from the training set. For instance, a learned witnesses may capture certain higher-level features of the training data. This related to "inductive bias introduced by the witness set" as suggested by the reviewer.
>
> It would certainly be interesting to investigate this phenomenon in more detail in future work.

---

### Official Review · Reviewer_eksT · 2025-07-01

**Clarity:** 2
**Significance:** 3
**Originality:** 3
**Rating:** 4
**Confidence:** 3

**Summary:**

This paper proposes to theoretically understand Transformers for in-context and out-of-context diffusion denoising. The tasks include in-context denoising, score-based diffusion denoising, denoising with learnable witness tokens, and anisotropic diffusion.  Through theoretical proof and experiments on MNIST and CIFAR-10, the paper proves that Transformers can perform iterative denoising for various denoising algorithms.

**Questions:**

Please refer to the Weaknesses.

**Ethical Concerns:**

["NO or VERY MINOR ethics concerns only"]

**Final Justification:**

Most of the concerns are addressed. I hope the author can organize the content in the rebuttal into the paper. I will keep my rating.

**Limitations:**

yes

**Paper Formatting Concerns:**

The paper is using a NeurIPS 2024 submission format.

**Quality:**

3

**Strengths And Weaknesses:**

### Strengths
- The paper is well-motivated, as performing iterative denoising across layers in a Transformer remains an underexplored area. This work provides valuable insights that could inspire future research.
- The paper provides solid theoretical derivations for various diffusion denoising tasks, which are supported by experimental results to justify the conclusions.

### Weaknesses
- The main concern is the practical applicability of the paper. While the paper offers theoretical analysis supporting the use of Transformers for diffusion denoising, **1)** it lacks a quantitative comparison with mainstream methods such as DiT [1] on more practical image generation tasks (such as benchmarks like COCO, ImageNet) and **2)** does not sufficiently discuss the conditions under which Transformers are more suitable than mainstream methods.
- An important aspect of modern denoising-based image generation methods is the flexibility to **1)** adjust the number of denoising steps to trade off between efficiency and quality, and **2)** to incorporate conditioning information such as text prompts. It would be valuable for the authors to discuss how their proposed method could support or be extended to enable these capabilities.
- The authors only conduct experiments using a six-layer Transformer and relatively simple benchmarks such as MNIST and CIFAR-10, which raises concerns about the scalability and generalizability of the proposed method to larger models and more complex datasets.
- The related work could be better organized into a section to provide a clearer understanding of the related tasks and contextualize the contributions of this paper more effectively.

[1] Scalable Diffusion Models with Transformers. ICCV 2023.

---

> ### Author Rebuttal · Authors · 2025-07-31
>
> We thank the reviewer for their thoughtful and comments and insightful questions. Below, we address each of the key points in turn.
>
> ## 1. Practical Applicability
> > The main concern is the practical applicability of the paper. While the paper offers theoretical analysis supporting the use of Transformers for diffusion denoising, 1) it lacks a quantitative comparison with mainstream methods such as DiT [1] on more practical image generation tasks (such as benchmarks like COCO, ImageNet) and 2) does not sufficiently discuss the conditions under which Transformers are more suitable than mainstream methods.
> > [1] Scalable Diffusion Models with Transformers. ICCV 2023.
>
> We have extended our experiments to the CIFAR100 dataset (requested by reviewer YpJ6), which we provide below. The loss values are very similar to CIFAR10. We aim to include more diverse and complex experiments (such as ImageNet, as well as other modalities) in the future draft.
>
> ### CIFAR10/CIFAR100 RMSE-vs-Layer
> | Layer | CIFAR-10 (Theory) | CIFAR-10 (Trained) | CIFAR-100 (Theory) | CIFAR-100 (Trained) |
> | ----- | ----------------- | ------------------ | ------------------ | ------------------- |
> | 1     | 30.9              | 20.03 ± 0.10       | 30.9               | 20.05 ± 0.08        |
> | 2     | 18.3              | 10.98 ± 0.02       | 18.3               | 10.93 ± 0.04        |
> | 3     | 12.6              | 9.98 ± 0.02        | 12.6               | 9.91 ± 0.04         |
> | 4     | 10.5              | 10.05 ± 0.03       | 10.6               | 9.97 ± 0.04         |
> | 5     | 10.0              | 10.12 ± 0.03       | 10.1               | 10.04 ± 0.04        |
> | 6     | 9.98              | 10.15 ± 0.03       | 10.0               | 10.06 ± 0.04        |
>
> ## 2. Flexible Denoising
> > An important aspect of modern denoising-based image generation methods is the flexibility to ... adjust the number of denoising steps to trade off between efficiency and quality
>
> We agree this is an important aspect of modern denoising-based generative models. Our approach naturally lends itself to this flexibility in two ways:
>
> - **Layerwise early stopping**: In Lemma 4, we show that each attention layer implements one denoising step. Figures 2 and 3 empirically demonstrate a monotonic decrease in loss with increasing layers. Thus, the model's output at an intermediate layer could be used as a lower-cost, lower-fidelity sample, this provides a direct mechanism to trade-off generation quality for speed.
> - A multi-step denoising process could be implemented by a *looped transformer* [1], in which the same attention parameters are applied recurrently across layers. Our Transformer construction in Lemma 4 uses the same weight configuration (up to scaling) for all attention layers, and thus extends naturally to this setting without modification. This approach **allows for scaling to many denoising steps with a small number of recurrent attention layers**. Furthermore, as recently explored in [2], test-time recurrence depth can be adapted based on input complexity—suggesting that **adaptive denoising steps** is feasible in our framework as well.
>
> [1] Can Looped Transformers Learn to Implement Multi-step Gradient Descent for In-context Learning? *Gatmiry, Nikunj Saunshi, Sashank J. Reddi, Stefanie Jegelka, Sanjiv Kumar 2024*
>
> [2] Scaling up Test-Time Compute with Latent Reasoning: A Recurrent Depth Approach *Jonas Geiping, Sean McLeish, Neel Jain, John Kirchenbauer, Siddharth Singh, Brian R. Bartoldson, Bhavya Kailkhura, Abhinav Bhatele, Tom Goldstein 2025*
>
> ## 3. Incorporating conditional information
> > An important aspect of modern denoising-based image generation methods is the flexibility to  ... incorporate conditioning information such as text prompts. It would be valuable for the authors to discuss how their proposed method could support or be extended to enable these capabilities.
>
> This is a very valuable suggestion. In text-to-image diffusion models, conditioning is typically achieved through cross-attention over textual embeddings. Our framework could naturally extend to this setting:
> 1. Both Lemma 4 and Lemma 5 can be modified to incorporate conditioning via cross-attention.
> 2. We point out a connection of the above to the "in-context score-based denoising" problem on line 200 of the paper. In this setup, the "conditioning information" is in the form of "contextual demonstration images". The Transformer denoises the query noise, conditioned on contextual demonstrations. This "conditioning on context" is achieved by having the query token attend to the demonstration tokens in the self-attention module.
>
> ## 4. Scalability and Generalizability
> > - The authors only conduct experiments using a six-layer Transformer and relatively simple benchmarks such as MNIST and CIFAR-10, which raises concerns about the scalability and generalizability of the proposed method to larger models and more complex datasets.
>
> We acknowledge that the architecture and datasets used in our experiments are relatively simple. However, this choice is deliberate: our primary goal is to provide a *mechanistic understanding* of how attention layers alone can implement in-context denoising.
>
> To isolate the role of attention, our architecture processes images as a single flattened pixel vector, *without any positional encodings or convolutional components*. Consequently, **the observed denoising behavior can be attributed entirely to the attention mechanism, independent of spatial inductive biases.** To our knowledge, this has not been systematically investigated in prior work.
>
> We agree that evaluating scalability to more complex datasets and architectures is an important future direction. In particular, adapting our constructions to settings involving positional encodings or patch-based tokenization (as in diffusion Transformers) will help illuminate how architectural components interact with the attention-driven mechanisms we uncover.
>
> > - The related work could be better organized into a section to provide a clearer understanding of the related tasks and contextualize the contributions of this paper more effectively.
>
> We thank the reviewer for their suggestion. We will include a detailed section on relation between our work, and prior work on 1) in-context learning, 2) score-based diffusion, and 3) manifold denoising.

---

> > ### Comment · Reviewer_eksT · 2025-08-04
> >
> > Thanks to the authors for their effort in the rebuttal. Most of the concerns were addressed. Still, I hope a comparison with DiT-liked structures can be provided. I will hold my original score currently.

---

> > > ### Author Response · Authors · 2025-08-05
> > >
> > > We thank the reviewer again for their thoughtful feedback and constructive suggestions. We appreciate the recommendation to compare with mainstream architectures such as DiT, and we will aim to include such comparisons in future versions of this work.

---

> > > > ### Comment · Reviewer_eksT · 2025-08-07
> > > >
> > > > Thanks to the authors for their effort in the rebuttal. Please also note that you are using the 2024 template. Good luck.

---

### Official Review · Reviewer_ukLB · 2025-07-01

**Clarity:** 1
**Significance:** 2
**Originality:** 3
**Rating:** 4
**Confidence:** 3

**Summary:**

This paper shows that transformers can perform denoising using in-context examples. Some theory is developed around this idea and experiments confirm the theory.

**Questions:**

- “uniform distribution over a (large) training set of samples” I am not sure I understand this one. Why do we care more about the uniform distribution over a set of examples as opposed to the true underlying signal distribution? In particular if we want to model the uniform distribution over a set of examples aren’t better equipped with simply sampling from it rather than using a diffusion model?
- What does “We verify that” mean  (line 148)?

**Ethical Concerns:**

["NO or VERY MINOR ethics concerns only"]

**Final Justification:**

Theoretical and experimental setup issues were clarified in the discussion.

Score is still pending on seeing the code as well as a plan for improving the clarity of the paper.

**Limitations:**

Yes

**Quality:**

2

**Strengths And Weaknesses:**

**Strengths**:
- The idea of using transformers for in-context learning to do denoising is really original and should be pushed further.

**Weaknesses**:
- *Code*: code is absent from the submission without justification. This greatly hinders reproducibility and reviewer’s ability to dive into the experimental setup.
- *Experimental setup and claims*: For each experiment it’s not clear what its purpose is in relationship to the theory laid out just before. For example it’s written:
> Figure 2a shows that the test loss decreases with the context length; this agrees with the fact that the discrete RBF Laplacian L becomes a better approximation to the Laplace Beltrami operator with increasing number of samples.

But I don’t understand how that relates to Lemma 3. nor why the the loss is the right thing to look at.
- *Assumption not discussed*: “We assume that smax masks out the diagonal entries of the similarity matrix” why does this make sense in this context? How does it deviate from typical practice? Normalization operations not presented in section 2, what difference does it make?
- *Phrasing/typos*: “ We highlight that the difference between the standard Attnstd and Attnrbf as follows”
- *Equivalence between Attnrbf and Attnrbf on sphere*: the lemma title is a bit misleading as it hides the fact that Q and K weight matrices also have to identity times a scalar.
- *Notations*: section 2 is extremely notation-heavy which makes it hard to follow.
- *Transformer and Kernel Weighted Update practical relevance*: it is not mentioned in this section what is the practical relevance of the kernel weighted update. Indeed there is no connection to prior literature using these kinds of transformers or illustration that these are useful in practice or are good approximations. On the other hand it seems exactly designed to fulfill the manifold denoising. A similar note can be said about the unconventional diffusion score modeling setup where the distribution we learn is a convoluted set of diracs.
- *Checklist*: the checklist has items marked as Yes or No without explanation, for example code or statistical significance.

---

> ### Author Rebuttal · Authors · 2025-07-31
>
> We thank the reviewer for their thoughtful comments and suggestions. Below, we address the main concerns, organized thematically for clarity.
>
> ## 1. Unconventional Diffusion Score Modelling Setup
>
> > “uniform distribution over a (large) training set of samples” I am not sure I understand this one. Why do we care more about the uniform distribution over a set of examples as opposed to the true underlying signal distribution? In particular if we want to model the uniform distribution over a set of examples aren’t better equipped with simply sampling from it rather than using a diffusion model?
>
> > ... A similar note can be said about the unconventional diffusion score modeling setup where the distribution we learn is a convoluted set of diracs
>
> We thank the reviewer for raising this point. We would like to clarify that our score modeling setup is **not unconventional**—it follows the standard practice in diffusion-based generative modeling.
>
> As described in lines 137–145 of our paper, our setup corresponds exactly to the **standard formulation of denoising score matching used in nearly all modern diffusion models.** Specifically, the empirical training distribution is **a uniform mixture of Dirac measures at the training samples**, convolved with a Gaussian kernel to define the perturbed data distribution \$p_t\$. This “convoluted set of Diracs” is exactly what appears in standard implementations of score-based models such as DDPM [1] and EDM [3].
>
> Because the true data distribution is unknown in practice, the training procedure *necessarily relies on the empirical distribution*. This is **consistent across virtually all generative modeling pipelines**. As noted by the reviewer, if we had access to the population distribution, *there would be no need for generative modeling at all.*
>
> The theoretical justification for learning scores on this empirical mixture is well-established. For instance, Appendix B.3 of [3] shows that in the infinite-capacity limit, a model trained on this empirical score-matching objective will converge to the true score of the empirical distribution \$p_t\$. The fact that practical models generalize beyond this set of Diracs is due to **architectural inductive biases, such as convolutional structure in UNets or tokenization in DiTs**. Analyzing the generalization properties of these inductive biases is an important research question, but one that is beyond the scope of our paper.
>
>
> ## 2. Relevance of Kernel Weighted Update
> > - Transformer and Kernel Weighted Update practical relevance: it is not mentioned in this section what is the practical relevance of the kernel weighted update. Indeed there is no connection to prior literature using these kinds of transformers or illustration that these are useful in practice or are good approximations. On the other hand it seems exactly designed to fulfill the manifold denoising.
>
> We thank the reviewer for this question. We would like to clarify both the theoretical and practical relevance of the Kernel-Weighted Update (KWU) framework.
>
> First, the KWU abstraction captures **both manifold-denoising [10], and score-based denoising [1]**. These two algorithms are very commonly used in practice. An important contribution of our paper is to show that KWU **unifies both paradigms under a single update rule**, thereby providing a common lens through which to interpret diverse denoising behaviors.
>
> Importantly, we prove in Lemmas 2–4 that **Transformers can implement KWU updates**, using standard attention layers. This immediately implies that Transformers have the capacity to simulate both manifold denoising and diffusion denoising dynamics.
>
> As for the practical relevance: the Transformer architecture we study **is conventional**. The only deviation from standard practice is the use of RBF attention (${Attn}_{\text{rbf}}$) instead of softmax attention. However, this difference is minor for the following reasons:
> - **we do show in Lemma 1 that these \$Attn_{rbf}\$ and \${Attn}_{smax}$ are exactly identical for inputs of constant norm**, which is almost always the case in practice due to the use LayerNorm in nearly all modern Transformers.
> - We also show in Lemma 8 that the the vectors in the denoising setup **strongly concentrate to a sphere of constant radius**.
> - Empirically, our experiments find that models trained with either attention **behave similarly**.
>
>
> ## 3. Relevance of Experiments
>
> > - Experimental setup and claims: For each experiment it’s not clear what its purpose is in relationship to the theory laid out just before. For example it’s written:
> >
> >>    Figure 2a shows that the test loss decreases with the context length; this agrees with the fact that the discrete RBF Laplacian L becomes a better approximation to the Laplace Beltrami operator with increasing number of samples.
> > But I don’t understand how that relates to Lemma 3.
>
> Lemma 3 shows that the Transformer is capable of implementing the denoising algorithm of [10], which is based on the Laplacian-based flow. Figure 2a is relevant for two reasons
> 1. As the context length increases, the discrete Laplacian approaches the Laplace Beltrami Operator (see e.g. Theorem 1 of [10]). Lemma 3 shows that the Transformer can simulate the discrete manifold flow; **as context size increases, this discrete flow converges to the continuous Laplace–Beltrami dynamics**. We explain this on lines 109-112,and will add a more detailed explanation in the next draft.
> 2. An important question in the in-context-learning literature is how the ICL loss scales with context length (e.g. [4])
>
> > nor why the the loss is the right thing to look at.
>
> The goal of in-context denoising is to recover \$x^{(i)}\$ given a noisy observation \$z^{(i)} = x^{(i)}+\epsilon\$. The L2 Euclidean distance between the denoised output and \$x^{(i)}\$ is a natural measure of recovery error. One could of course also consider alternatives like L1 loss.
>
> **Please let us know if you would like an explanation of the purpose of the other experiments.**
>
> ## 4. Transformer Architecture Questions
>
> > - Assumption not discussed: “We assume that smax masks out the diagonal entries of the similarity matrix” why does this make sense in this context? How does it deviate from typical practice?
>
> "Masking out diagonal entries of the similarity matrix" means that each token does not attend to itself. It is a very common setup, and is implied, for instance, by lower-triangular mask matrices which are used in causal self-attention. It is also common to enforce this in manifold denoising, see e.g. Section 3.1 of [10].
>
>
> > - Equivalence between Attnrbf and Attnrbf on sphere: the lemma title is a bit misleading as it hides the fact that Q and K weight matrices also have to identity times a scalar.
>
> We do state that \$W^Q\$ and \$W^K\$ are scalings of identity in the sentence on the second line of Lemma 1 (line 86). Additionally, \$Attn_{rbf}\$ and \$Attn_{std}\$ are functions of vector embeddings. The title highlights the fact that input vector embeddings must lie on the sphere.
>
> > Normalization operations not presented in section 2, what difference does it make?
> - If you mean "normalization to constant norm for tokens", that would be exactly LayerNorm, which is commonly used in Transformers. In addition, Lemma 8 provides a concentration bound for the radius \$x_t \sim p_t\$ around \$\sqrt{td}\$, assuming \$p_0\$ is normalized.
> - If you mean "normalization of probability distribution", that is a natural part of softmax, see e.g. proof of Lemma 1 in Appendix A.
> - If you are referring to some other normalization, please clarify and we would be happy to follow up.
>
> ## 5. Miscellaneous Issues
>
> > - Notations: section 2 is extremely notation-heavy which makes it hard to follow.
>
> We will add more exposition to make the section easier to follow.
>
> > - What does “We verify that” mean (line 148)?
>
> It means that we can verify that the continuity equation (9) describes the density evolution in (8). There are several ways to verify this, such as via the Fokker Planck equation. We can add a more detailed proof in the appendix in the next draft. The proof can also be found in many existing diffusion model papers such as [2].
>
>
> > - Code: code is absent from the submission without justification. This greatly hinders reproducibility and reviewer’s ability to dive into the experimental setup.
>
> We have uploaded the code as an anonymized link to the AC in an Official Comment (per the author FAQ). We believe that the AC provide the link to you soon.
>
> > - Checklist: the checklist has items marked as Yes or No without explanation, for example code or statistical significance.
>
> We will include std bounds for all data in the next draft of our paper. The standard deviation is generally significantly less than \$1\%\$ of the mean value. We provide below the std bounds for Figure 3 as an example. Please let us know if you would like the STD bounds for data in the other figures.
>
> | Layer | RBF Trained           | STD Trained           |
> |-------|------------------------|------------------------|
> | 1     | 20.0300 ± 0.0997       | 30.8837 ± 0.0031       |
> | 2     | 10.9777 ± 0.0206       | 18.2155 ± 0.0091       |
> | 3     | 9.9822 ± 0.0233        | 12.4484 ± 0.0185       |
> | 4     | 10.0486 ± 0.0253       | 10.3784 ± 0.0260       |
> | 5     | 10.1176 ± 0.0257       | 9.8413 ± 0.0297        |
> | 6     | 10.1461 ± 0.0257       | 9.7701 ± 0.0311        |
>
> *\*RBF Theory and STD Theory have 0 variance.*
>
> ## References
>
> [1] Denoising Diffusion Probabilistic Models, *Jonathan Ho, Ajay Jain, Pieter Abbeel*
>
> [2] Score-Based Generative Modeling through Stochastic Differential Equations, *Song et el 2020*.
>
> [3] Elucidating the Design Space of Diffusion-Based Generative Models, *Karras et al 2022*
>
> [4] What learning algorithm is in-context learning? Investigations with linear models, *Akyurek et el 2023*
>
> [10] Manifold Denoising, *Matthias Hein, Markus Maier 2006*.

---

> > ### Comment · Reviewer_ukLB · 2025-08-03
> >
> > I would like to thank the authors for engaging in the review process in a constructive and respectful manner.
> >
> > **Unconventional Diffusion Score Modelling Setup**: While the EDM paper [1] provides a derivation in the case of a finite dataset in their Appendix 3., their goal is rather to learn the true underlying data distribution just like the other classical generative modeling efforts. If we focus on diffusion works for example:
> > Song et al. [2]  say “Suppose our dataset consists of i.i.d. samples ${x_i \in \mathbb{R}^D}^N_{i=1}$ from an unknown data distribution $p_\text{data}(x)$.”. The data distribution is unknown, not a set of diracs.
> > Ho et al. [3] make no assumptions that the data distribution they are trying to model should be a set of diracs.
> >
> > I think there probably was a confusion about the point I was trying to make. Obviously, when doing generative modeling we have access to a finite set of samples of the true underlying data distribution, I am not trying to dispute that. My only point is that we are actually trying to model (sample from) the true underlying data generative distribution as opposed to the empirical distribution given by the set of diracs over the samples.
> >
> > “As noted by the reviewer, if we had access to the population distribution, there would be no need for generative modeling at all.” -> I didn’t write this in my original review. What I said was “if we want to model the uniform distribution over a set of examples aren’t better equipped with simply sampling from it rather than using a diffusion model?” which was indeed saying that if we want to sample from the empirical distribution, the uniform distribution over the set of examples is sufficient. This was a way to say that what we want to model (sample from) is the true underlying distribution.
> >
> > “The fact that practical models generalize beyond this set of Diracs is due to architectural inductive biases, such as convolutional structure in UNets or tokenization in DiTs.” -> Indeed practical models because they are not infinite-capacity cannot match the score of the empirical distribution perfectly and I agree that this is why even in the finite-data regime we do not have issues with mode collapse. However, the justification is not solely based on unknown inductive biases, but also on the fact that the expectations over the true data distribution are approximated by the expectations over the empirical distribution and that they asymptotically match.
> >
> > **Relevance of Kernel Weighted Update**: After re-reading the claims in the paper and the rebuttal on this point I understand better the context. I agree that the paper makes it clear that transformers with some modifications can implement kernel weighted updates. I think I was confused by the title of the paper which claims “Transformers are Effective Diffusion Denoisers, Both in Context, and Without Context.” and not that they *can* be effective denoisers. To me the difference is actually very important because we know that transformers are very powerful function approximators. Therefore it is not clear to me why the specific contributions of this paper of showing what kind of algorithms the transformers can approximate (given some non trivial modifications, like forcing some weight matrices to be identity times a scalar) are not special cases of function approximation. I may be missing something obvious and this is a very genuine question.
> >
> > **Relevance of Experiments**:
> > Lines 109-112 of the manuscript read:
> > > This problem is particularly challenging because (1) the manifold Mis unobserved, and (2)
> > the manifold Mis context-dependent. Therefore, to solve this problem in-context, the Transformer
> > needs to be able to learn (implicitly or explicitly) the manifold structure, from the observations x(i)’s.
> > [10] propose the following manifold-denoising algorithm based on the Laplace-Beltrami operator:
> >
> > I don’t think it explains that “as context size increases, this discrete flow converges to the continuous Laplace–Beltrami dynamics” as mentioned in the response.
> > “will add a more detailed explanation in the next draft.” -> I think it would be best to have it in the response so the reviewers can judge appropriately.
> > I think my original point was really to say -> there is nothing in the lemma or in a follow-up that says “we therefore expect that when training a transformer on … we will see the loss go down with context size.” and then later the experiment shows it.
> >
> > **Miscellaneous Issues**:
> > “We have uploaded the code as an anonymized link to the AC in an Official Comment” -> Makes sense, I have not yet received this link and am waiting for it.
> >
> >
> > [1] Elucidating the Design Space of Diffusion-Based Generative Models, Karras et al 2022
> > [2] Song, Y., & Ermon, S. (2019). Generative modeling by estimating gradients of the data distribution. Advances in neural information processing systems, 32.
> > [3] Denoising Diffusion Probabilistic Models, Jonathan Ho, Ajay Jain, Pieter Abbeel

---

> > > ### Author Response · Authors · 2025-08-04
> > >
> > > ## Unconventional Diffusion Score Modelling Setup
> > >
> > > We thank the reviewer for the thoughtful clarification. To ensure alignment, we summarize what we believe is common ground:
> > >
> > > 1. The ultimate goal of generative modeling is to sample from the true data distribution $p_{\text{data}}$, not the empirical distribution $p_{\text{train}}$.
> > > 2. However, in practice—as in \[1,2,3] and virtually all modern diffusion models—the training loss is defined with respect to $p_{\text{train}}$, the uniform mixture of Dirac masses at the training samples.
> > > 3. The optimal unconstrained denoiser (or velocity field) learned from this loss corresponds to the score of the convolved distribution $p_t = p_{\text{train}} * \mathcal{N}(0, tI)$.
> > > 4. Consequently, exactly implementing the reverse ODE using this score will generate samples from $p_{\text{train}}$, not $p_{\text{data}}$.
> > >
> > > The training loss, theoretical constructions, and the way we implement our experiments are all defined with respect to $p_{\text{train}}$, *just like in standard score-based diffusion*. This setup—including all of our results—remains **unchanged regardless of whether the intended modeling goal is to sample from $p_{\text{train}}$ or from $p_{\text{data}}$**. This mirrors the standard diffusion setting, where the training loss is always over $p_{\text{train}}$, even though the aspirational target is often $p_{\text{data}}$.
> > >
> > > That said, we emphasize that our focus is indeed on generalization to $p_{\text{data}}$. Specifically:
> > >
> > > * All losses reported in Figures 2, 3, 5, and 7 are computed on **unseen test examples**, not the training set.
> > > * As with score-based models, finite-capacity architectures—especially those with structural constraints, like our witness-based Transformer (Figure 7)—can and do generalize beyond $p_{\text{train}}$.
> > >
> > > **Therefore, we believe the reviewer’s concern may be primarily about presentation**. If so, we agree—our current draft could more clearly state that while training is defined over $p_{\text{train}}$, our modeling goal is generalization to $p_{\text{data}}$. We will include this clarification in the next revision.
> > >
> > > **If instead the concern is conceptual**—i.e., that our framework only applies when the modeling goal is to sample from $p_{\text{train}}$—we would like to further clarify:
> > >
> > > 1. **None of our theory or experiments rely on any assumption** that the true data distribution $p_{\text{data}}$ is a sum of Diracs.
> > > 2. Regardless of training set size, the **optimal infinite-capacity model** trained under our setup (as well as under the standard score-matching setup) **will only generate samples from the training set**.
> > > 3. **For finite-capacity models, both our setup and standard score-matching can generalize beyond the training set.** In our paper, the witness-based Transformer in Figure 7 produces outputs distinct from any training example and thus generalizes better than the baseline (lower test loss).
> > >
> > > (continued below)

---

> > > > ### Author Response · Authors · 2025-08-04
> > > >
> > > > ## Relevance of Kernel Weighted Update
> > > > > To me the difference is actually very important because we know that transformers are very powerful function approximators. Therefore it is not clear to me why the specific contributions of this paper of showing what kind of algorithms the transformers can approximate (given some non trivial modifications, like forcing some weight matrices to be identity times a scalar) are not special cases of function approximation.
> > > >
> > > > We thank the reviewer for this important question, as it allows us to clarify the core contribution of our work. While it is true that Transformers are universal function approximators, our theoretical and empirical results go significantly beyond universal function approximation:
> > > >
> > > > 1. **Universal approximation results are existential and non-constructive**: they guarantee approximation in principle, but often require impractically large networks. For example, while a wide two-layer ReLU network can approximate any function, such a network is likely **unable to implement score-based denoising with reasonable size or interpretability, even if the weights were hand-crafted.**
> > > > 2. In contrast, our constructions are **explicit, simple, and interpretable**. We show that a single-head attention layer with weights constrained to be identity-scaled already implements a meaningful kernel-weighted update (Lemma 2). The model doesn’t merely approximate the final input–output map of a denoising algorithm—it executes the algorithm step-by-step, with each attention layer corresponding exactly to one denoising iteration.
> > > > 3. This provides a powerful explanatory lens. It suggests a reason *why* Transformers are effective at these tasks: their core architectural component (attention) shares a fundamental mathematical structure with principled denoising algorithms.
> > > > 4. The modifications we make apply only to **how the Transformer is trained**—specifically, by constraining weights such as $W^Q$ and $W^K$ to be scaled identity matrices. These constraints make optimization and analysis tractable, but they are **not architectural modifications**: the resulting weights, whether constrained or learned freely, can be implemented in a **standard Transformer with no changes to the attention mechanism**. Aside from the use of RBF instead of softmax attention (which we show to be interchangeable under standard normalization), our constructions in Lemmas 2–4 can be realized in a standard Transformer.
> > > > 5. Beyond these results on the *algorithmic capacity* of the Transformer, we agree that an important direction is to understand what a full Transformer actually learns during training. We take preliminary steps toward this in Section 4.4: **we show that when $W^Q, W^K, W^V$ are allowed to be general learnable diagonal matrices, the resulting attention layer implements an anisotropic, geometry-aware denoising process**. This extends beyond the identity-based construction, and demonstrates that Transformers can *learn* more powerful denoising algorithms. We fully characterize this richer family of anisotropic denoisers and show empirically in Figure 7 that they outperform their isotropic counterparts.
> > > > In summary, our contribution is not one of function approximation, but of providing a precise, constructive, and interpretable link between the mechanism of attention and the process of iterative denoising.
> > > >
> > > > >  I think I was confused by the title of the paper which claims “Transformers are Effective Diffusion Denoisers, Both in Context, and Without Context.” and not that they can be effective denoisers
> > > >
> > > > The reviewer's point about the title is well-taken. We would be happy to revise the title to something more precise, such as "**Transformers as Effective Diffusion Denoisers...**" or "**On the Capacity of Transformers to Act as Diffusion Denoisers...**" to avoid ambiguity.
> > > >
> > > > (continued below)

---

> > > > > ### Author Response · Authors · 2025-08-04
> > > > >
> > > > > ## Relevance of Experiments and Connection to Theory
> > > > >
> > > > > > ...I think my original point was really to say -> there is nothing in the lemma or in a follow-up that says “we therefore expect that when training a transformer on … we will see the loss go down with context size.” and then later the experiment shows it.
> > > > >
> > > > > We apologize that the connection between Lemma 3 and the experiment in Figure 2a was not made explicit enough. We thank the reviewer for the opportunity to clarify our reasoning, which we will add to the revised manuscript.
> > > > >
> > > > > The logical chain is as follows:
> > > > > 1.  **Our Theory (Lemma 3):** We prove that a Transformer layer can exactly implement one step of the discrete manifold denoising algorithm from [10], which is based on the discrete Laplacian $\mathcal{L}$. This operator is constructed from the $n$ in-context samples.
> > > > > 2.  **Known Results in Manifold Learning:**  (Hein & Maier 2006 [10]):
> > > > >     - Theorem 1 of [10] establishes that as **as $n$ goes to $\infty$**, the discrete Laplacian $\mathcal{L}$ converges to the continuous Laplace-Beltrami operator $\Delta_\mathcal{M}$ on the underlying manifold.
> > > > >     - Lemma 2 of [10] uses Theorem 1 to show that the discrete denoising algorithm converges to an ideal flow, based on $\Delta_\mathcal{M}$ (mean-curvature flow + contraction towards manifold projection).
> > > > > 3.  **Hypothesis:** As the number of context points $n$ increases, the Transformer implements a discrete flow that more accurately approximates this ideal flow based on $\Delta_{\mathcal{M}}$ on the true data manifold.
> > > > > 5.  **Experimental Validation:** A more accurate denoising process should result in a lower error when reconstructing the clean signal from the noisy one. The L2 loss directly measures this denoising error. Figure 2a empirically tests this hypothesis by showing that the test loss (error) indeed decreases as the context length $n$ increases.
> > > > >
> > > > > This confirms the entire chain of reasoning: our Transformer implements a discrete algorithm (Lemma 3), which better approximates the ideal algorithm as context size grows (manifold learning theory), leading to lower error (Figure 2a). We will explicitly add this detailed explanation to the paper.
> > > > >
> > > > >
> > > > > ## Miscellaneous Issues
> > > > >
> > > > > We thank the reviewer for their patience regarding the code. We have double-checked that the anonymized link was indeed posted to the Area Chair last week.
> > > > >
> > > > > We thank the reviewer's detailed questions. We hope these clarifications adequately address the remaining concerns.

---

> > > > > > ### Comment · Reviewer_ukLB · 2025-08-04
> > > > > >
> > > > > > Ack'ed! Thanks for taking the time to explain well and meeting me halfway this well appreciated.
> > > > > >
> > > > > > Will now proceed to re-read the paper with these elements in mind, I think in the meantime it would be great to see an explicit plan for how to embed these explanations in the paper given the space constraints.

---

> ### Comment · Reviewer_ukLB · 2025-08-04
> **Raising score from 2 to 4 pending on explicit plan to improve clarity**
>
> After a re-read of the paper and given the context of the reviewer-author discussion here are some points I would like to discuss.
>
> - **Context**: the question asked by the paper is: “How do attention layers collectively implement a stepwise denoising process—a generative flow through high-dimensional space?”. First I think with the context of the discussion it should probably be “How could”. Second, I think there is a bit of a lack of context as to why we are asking ourselves this question. I understand that there are some other works looking at the capabilities of transformers in the forward pass but why denoising specifically, why do we think transformers could be useful in that context, or rather why do we think in-context learning could be useful?
> - **Denoising order**: “Notice that each Transformer layer denoises a roughly non-overlapping patch” -> is there an intuitive explanation for this?
> - **Title**: “We would be happy to revise the title to something more precise” -> I think the propositions make sense
> - **Relevance of Experiments and Connection to Theory**: ok so the way I understand it now, is “we proved that transformers can implement this algorithm which by the way has property A, and look when we tweak the weights of the transformers to make it implement said algorithm it showcases property A”. I guess it makes sense.
> - **Unconventional Diffusion Score Modelling Setup**: I guess indeed the concern is more about presentation.
> - **Relevance of Kernel Weighted Update**: I think the explanation in the response makes total sense. One thing I would like perspective on (and I guess this ties back to context as well) is how would we connect these results then to practical applications or in other words what do we expect to gain from this novel understanding?
>
>
> I think provided a solid explicit plan (i.e. in the response in full, with explicit rewrites) for how to integrate the clarifications from the discussion there are enough reasons to accept this paper.
> I will raise my score accordingly from 2 to 4, again pending on a solid explicit plan to improve clarity and context of the paper.
>
> I would like to underline the quality of the authors’ response and discussion which was really well thought-through.

---

> ### Author Response · Authors · 2025-08-05
> **Plan for incorporating reviewer discussion (1)**
>
> We thank the reviewer for the thoughtful and constructive feedback throughout the discussion period. Your comments have helped us significantly improve the clarity, motivation, and framing of the paper. In response, we have made several explicit edits to the manuscript, which are organized into five main categories below. In particular, the latest discussion points—regarding **context and motivation**, the order of anisotropic denoising, and practical implications—are now addressed directly in the revised text (under Category 1, Category 2 and Category 1 respectively).
>
> ## 1. Clarifying Context, Motivation and Practical Motivation
>
> We will merge the following paragraph with lines 33-37 in the Introduction:
> > **Motivation**. Denoising is a fundamental operation in modern generative modeling. It lies at the heart of diffusion-based models and is a canonical example of unsupervised learning, where structure must be recovered from corrupted or incomplete observations. At the same time, recent theoretical work on in-context learning (ICL) has focused almost exclusively on supervised tasks, where clean input-label pairs are available. However, contexts encountered in practical settings—both in vision and language—are often noisy, unlabeled, or partially observed. This motivates our investigation of unsupervised in-context learning, and specifically, whether Transformers can implement stepwise denoising flows during inference. Transformers have demonstrated strong empirical performance in generative tasks, but lack a principled interpretation as iterative denoisers. Our goal is to bridge this gap: we show that attention-based architectures not only support structured denoising updates, but can also learn geometry-aware anisotropic diffusion algorithms when trained end-to-end. This analysis provides algorithmic insight into why Transformers excel at generation and offers a unified lens for understanding both score-based diffusion and manifold learning as forms of Transformer-implemented denoising. Our work also connects to the emerging literature on in-context generation with diffusion models [11,12], highlighting the relevance of unsupervised ICL in this setting.
>
> We will also add a section "Practical Implications and Future Work" right before the "Limitations" section:
>
> > **Practical Implications.** Our results offer several concrete takeaways for the design and analysis of Transformer-based generative models.
> >
> > * **In-context generation.** Recent work on in-context generation—across both diffusion \[11] and autoregressive models \[12]—calls for a principled understanding of how Transformers incorporate contextual information. Our analysis provides such a framework: attention layers can naturally implement stepwise denoising conditioned on context. For instance, our theory offers a formal justification for using  *cross-attention* to attend to in-context samples during denoising—reducing computational complexity from quadratic to linear in context length, while preserving performance. This idea could benefit tasks like *few-shot image editing*, *personalized generation*, or *test-time adaptation*, where guidance from reference samples is critical.
> >
> > * **Parameter-efficient denoising.** Our characterization of anisotropic denoising with diagonal attention weights suggests that *even highly constrained parameterizations* can learn powerful denoising algorithms. This opens up opportunities for efficient Transformer variants in generative models—for example, using *low-rank or diagonal attention* in diffusion Transformers, or fine-tuning these components in parameter-efficient learning settings.
> >
> > * **Localized, interpretable attention.** We observe that Transformers often specialize different layers or heads to denoise *non-overlapping semantic patches* (Figure 1). This suggests a promising direction for designing *sparse, locality-aware attention mechanisms* in structured domains like vision. Such mechanisms may be more interpretable and robust to noisy or incomplete context.
>
> [11] In-Context Learning Unlocked for Diffusion Models, Wang et al 2023.
>
> [12] CoDi-2: In-Context Interleaved and Interactive Any-to-Any Generation, Tang et al 2024.
>
> (continued below)

---

> > ### Author Response · Authors · 2025-08-05
> > **Plan for incorporating reviewer discussion (2)**
> >
> > ## 2. Clarifying Problem Setup and Modelling Goal
> >
> > We will add the following paragraph to the start of Section 4 (Score-Based Denoising) to clarify the real goal of generating from $p_{data}$.
> >
> > >As in standard score-based diffusion models [1, 2, 3], our training objective is defined over the empirical distribution $p_{\text{train}}$, the uniform distribution over training samples ${x^{(1)}, \dots, x^{(n)}}$. We assume these samples are drawn i.i.d. from an unknown population distribution $p_{\text{data}}$.
> > >It is important to emphasize that while the training distribution is $p_{\text{train}}$, the modeling goal is to generalize to $p_{\text{data}}$. This distinction is shared across all score-based diffusion models: although the loss is minimized on finite training data, the goal is to sample from the broader population distribution. Our theoretical results and empirical setup are fully agnostic to properties of $p_{\text{data}}$.
> > >All evaluations (e.g., Figures 3, 5, and 7) are performed on unseen test samples.
> >
> > We will add the following discussion after Figure 7 to elaborate on generalization advantage of witness models:
> > > Remarkably, the witness model achieves lower test loss than the Baseline model, which implements exact score-based denoising. This highlights an important phenomenon: although the Baseline perfectly fits $p_{\text{train}}$, its lack of architectural constraints leads to poor generalization. In contrast, the Witness model’s restricted attention (which prevents attending to all training samples) and its learned inductive bias (via summary witnesses) enable it to generalize better to the unseen data distribution $p_{\text{data}}$.
> >
> > We will also add a brief note to the captions of Figures 2,3,5,7:
> > > We evaluate on test (unseen) data to assess generalization.

---

> ### Author Response · Authors · 2025-08-05
> **Plan for incorporating reviewer discussion (3)**
>
> ## 3. Clarify Kernel-Weighted Update (KWU) Contributions
>
> **Technical Contribution of KWU Transformer Construction**:
>
> We will change the title to
> > On the Capacity of Transformers as Effective Denoisers, Within and Without Context.
>
> We will add the following discussion following Lemma 2 (Transformer Construction for KWU Update) to motivate the KWU framework, and motivate our Transformer construction.
>
> >The KWU abstraction captures both manifold denoising (Section 3) and score-based diffusion denoising (Section 4). By unifying these paradigms under a common update rule, KWU provides a single lens through which to interpret a range of denoising behaviors. Showing that Transformers can implement KWU updates implies they have the capacity to simulate both manifold-based denoising (Lemma 3) and diffusion-based denoising dynamics (Lemma 4).
> >While Transformers are known to be universal function approximators, such results are non-constructive and often require large networks to approximate specific algorithms. In contrast, our contribution is simple, constructive, and interpretable. We show that a standard attention layer—with identity-scaled weights—can exactly implement a kernel-weighted denoising update (Lemma 2). This suggests a reason why diffusion Transformers are effective in practice: their core architectural mechanism, attention, mirrors the structure of principled denoising algorithms.
>
> **Anisotropic Denoising via Learnable Parameters**: We add the following discussion to the start of Section 4.4. It motivates the anisotropic-denoising setting, and emphasizes that we investigate *what Transformers learn* instead of *what Transformers can implement*.
>
> >Our constructions in Lemmas 2, 3, and 4 focus on the algorithmic capacity of the Transformer by demonstrating that it can implement isotropic denoising. A natural question is whether a fully learnable Transformer can realize richer, data-adaptive denoising behaviors.
> >In Section 4.4, we take preliminary steps to answer this. We show that when the attention weights $W^Q, W^K, W^V$ are learned diagonal matrices, the resulting update corresponds exactly to an *anisotropic, geometry-aware denoising process*. We provide a rigorous characterization of this richer family of anisotropic denoising algorithms (Lemma 5), and empirically verify that the learned anisotropic model outperform the isotropic baseline in test loss (Figure 7).
>
> **Non-overlapping Denoising Patches**: We add the following discussion to the last paragraph of Section 4.5 to provide an intuitive explanation for why Transformer leans to denoise in patches:
>
> > Remarkably, the anisotropic denoising process learned by the Transformer is highly interpretable. As shown in Figure 1, each layer of the Transformer appears to denoise a distinct, non-overlapping local patch of the image.
> >We conjecture that this behavior arises for two reasons:
> >1. **Exploiting local pixel correlations.** Consider the extreme case where all pixels within a patch have the same color. In this case, denoising a single pixel is equivalent to denoising the entire patch, making it more efficient to denoise the patch as a whole.
> >2. **Efficient parameter reuse.** Across a dataset, many images may contain visually similar patches. A small number of "summary witnesses" can specialize in denoising these recurring structures, allowing the Transformer to reuse and generalize denoising behavior efficiently.

---

> ### Author Response · Authors · 2025-08-05
> **Plan for incorporating reviewer discussion (4)**
>
> ## 4. Clarify Relevance of Experiments to Theory
>
> We add the following discussion to the last paragraph of Section 3.1, based on the suggestion by the reviewer.
>
> > Lemma 3 shows that a Transformer can implement a discrete manifold denoising update based on the graph Laplacian. This algorithm is known to converge to an ideal continuous flow on the data manifold as the number of context points increases—specifically, Theorem 1 and Lemma 2 of [10] show that the discrete Laplacian converges to the Laplace–Beltrami operator, and the resulting discrete flow converges to mean curvature flow. We empirically verify that the Transformer, which exactly implements this discrete algorithm, exhibits the same convergence behavior: as the number of context points increases, denoising performance improves (Figure 2a), validating this theoretical prediction.
>
> $ $
>
> ## 5. Improve Notation, Assumptions, and Presentation
>
> To address the reviewer’s helpful feedback on notation and architectural assumptions, we will revise the paper in the following ways:
>
> #### (a) Improve Section 2 for Notational Clarity
> * **Add a short prose lead-in** before the first equation to motivate the symbols.
> * **Introduce a notation table** at the end of Section 2 covering only the symbols that are used throughout the paper:
>
> | Symbol                                               | Meaning                                          | Shape                    |
> | ---------------------------------------------------- | ------------------------------------------------ | ------------------------ |
> | $z^{(i)}$                                            | $i^{\text{th}}$ input token                      | $\mathbb{R}^d$           |
> | $Z_0=[z^{(1)}\dots z^{(n)}]$                       | token matrix                                     | $\mathbb{R}^{d\times n}$ |
> | $W_\ell^V,W_\ell^Q,W_\ell^K,W_\ell^S$                | value, query, key, skip matrices at layer $\ell$ | $\mathbb{R}^{d\times d}$ |
> | $Attn_{std}, Attn_{rbf}$ | soft-max and RBF attention operators             | —                        |
> | $\kappa_\ell(i,j)$                                   | RBF kernel weight (Eq. 5)                        | scalar                   |
> * **Add a clarification on dimension conventions** so the reader need not infer them from context (tokens are column vectors, matrices are $d\times n$, etc.).
>
> These edits keep the core definitions ($z^{(i)}$, $Z_0$, $W_\ell^\cdot$, attention operators, kernel $\kappa_\ell$) visible while eliminating extraneous notation early on.
>
>
> #### (b) Clarify Diagonal Masking and Its Role in Self-Attention
>
> - The sentence:
>
>   > "We assume that softmax masks out the diagonal entries of the similarity matrix..."
>
>     will be revised and clarified as follows:
>   > "We use a diagonal mask to prevent each token from attending to itself. This is standard in causal self-attention and manifold denoising, as it ensures the denoising update at each position is informed by neighboring tokens rather than itself (see Section 3.1 of [10])."
>
> * A footnote or in-line clarification will explain that this practice enforces proper autoregressive or non-self denoising structure and is aligned with both Transformer and manifold learning conventions.
>
> #### \(c) Clarify Assumptions in Lemma 1
>
> * For **Lemma 1** (“Equivalence of Attn_{rbf} and Attn_{std} on the sphere”), we will revise the lemma title or annotate in-line to more explicitly emphasize the assumption:
>
>   > “under the assumption that $W^Q$ and $W^K$ are identity-scaled matrices.”
>
> #### (d) Clarify "We verify that..." (line 148)
>
> * We will clarify:
>
>   > “We verify that the continuity equation (9) governs the evolution of $p_t$ as described in (8). This can be derived from the Fokker-Planck equation, and we will include a formal derivation in Appendix A. Similar arguments appear in many diffusion model papers (e.g., \[2]).”

---

### Official Review · Reviewer_YpJ6 · 2025-07-03

**Clarity:** 3
**Significance:** 3
**Originality:** 3
**Rating:** 4
**Confidence:** 3

**Summary:**

This paper studies whether Transformer architectures can inherently perform diffusion-based denoising and draws connections between Transformer attention mechanisms and classical denoising algorithms. It presents a unified theoretical framework showing that Transformers can exactly implement various denoising processes: (a) manifold denoising via Laplacian smoothing flows, (b) score-based denoising as used in diffusion probabilistic models, and (c) anisotropic diffusion processes. The authors prove an equivalence between Transformer attention updates and these denoising algorithms under certain conditions, and introduce a method using learnable “witness” tokens to approximate the diffusion score more efficiently. They validate these insights on image denoising tasks, demonstrating that simple Transformers can perform robust iterative denoising both with additional context (in-context learning scenarios) and without context (standard generative diffusion). These results highlight the Transformer's flexibility as a denoising meta-learner, capable of bridging context-dependent and context-independent generative modeling.

**Questions:**

See weaknesses

**Ethical Concerns:**

["NO or VERY MINOR ethics concerns only"]

**Final Justification:**

Thanks for the rebuttal that addresses my concerns.

**Limitations:**

See weaknesses

**Quality:**

3

**Strengths And Weaknesses:**

Strengths

1. Provides proofs linking Transformer attention to manifold, score-based, and anisotropic diffusion denoising, offering a unified mathematical perspective that is novel and well-substantiated.

2. Controlled experiments on MNIST/CIFAR-10 show monotonic error reduction across layers, effective use of context, and clear benefits from learnable tokens, confirming the theoretical claims in practice.

3. Introduces learnable witness tokens and extends the framework to anisotropic diffusion, yielding significant efficiency gains and opening new avenues for Transformer-based generative modeling.

Weaknesses

1. Results are confined to small-scale vision datasets and shallow, single-head Transformers; there is no evidence the approach scales to high-resolution images, deeper models, or other modalities. Even under constrained compute budgets, it would strengthen the paper to report results on more additional small- or medium-scale benchmarks like CIFAR-100, Tiny ImageNet for images, or SpeechCommands and UrbanSound8K for audio, to demonstrate broader applicability.

2. The study reports only pixel-level RMSE, omitting perceptual metrics (e.g., SSIM, FID) and comparisons to state-of-the-art denoisers, so practical denoising quality and generative relevance remain uncertain.

---

> ### Author Rebuttal · Authors · 2025-07-31
>
> We thank the reviewer for their thoughtful comments and constructive suggestions for improving our experiments. We address the main concerns below.
>
> ## 1. Generalization beyond CIFAR-10 and Shallow Transformers
>
> > Results are confined to small-scale vision datasets and shallow, single-head Transformers; there is no evidence the approach scales to high-resolution images, deeper models, or other modalities. Even under constrained compute budgets, it would strengthen the paper to report results on more additional small- or medium-scale benchmarks like CIFAR-100, Tiny ImageNet for images, or SpeechCommands and UrbanSound8K for audio, to demonstrate broader applicability.
>
> We appreciate the reviewer’s suggestion and agree that demonstrating broader applicability is important. To that end, we include **new results on CIFAR-100** using the same denoising implementation described in Section 4. We report RMSE values for CIFAR-100 alongside the existing CIFAR-10 results below. As shown, **RMSE trends are highly consistent across both datasets**, suggesting that the observed denoising behavior is not specific to a particular dataset.
>
> We view this as a promising indication that our findings extend beyond CIFAR-10. We aim to include more complex/diverse datasets like Tiny Imagenet and UrbanSound8K in the future.
>
> ### CIFAR10/CIFAR100 RMSE-vs-Layer
> | Layer | CIFAR-10 (Theory) | CIFAR-10 (Trained) | CIFAR-100 (Theory) | CIFAR-100 (Trained) |
> | ----- | ----------------- | ------------------ | ------------------ | ------------------- |
> | 1     | 30.9              | 20.03 ± 0.10       | 30.9               | 20.05 ± 0.08        |
> | 2     | 18.3              | 10.98 ± 0.02       | 18.3               | 10.93 ± 0.04        |
> | 3     | 12.6              | 9.98 ± 0.02        | 12.6               | 9.91 ± 0.04         |
> | 4     | 10.5              | 10.05 ± 0.03       | 10.6               | 9.97 ± 0.04         |
> | 5     | 10.0              | 10.12 ± 0.03       | 10.1               | 10.04 ± 0.04        |
> | 6     | 9.98              | 10.15 ± 0.03       | 10.0               | 10.06 ± 0.04        |
>
>
> ## 2. Lack of Perceptual Metrics and Comparison to SOTA
> > The study reports only pixel-level RMSE, omitting perceptual metrics (e.g., SSIM, FID) and comparisons to state-of-the-art denoisers, so practical denoising quality and generative relevance remain uncertain.
>
> We agree with the reviewer that perceptual metrics offer valuable insights. In response, we have computed Fréchet Inception Distance (FID) for CIFAR-10 and CIFAR-100 as a function of Transformer depth. The results reported in the table below.
>
> As expected, the FID improves substantially with network depth and is consistent across datasets. We note that the absolute FID values are significantly higher than those of state-of-the-art generative models. This is expected due to the **highly minimalistic** nature of our architecture:
>
> 1. **No positional encoding**: our model does not possess spatial inductive bias.
> 2. **Flattened input**: no patching, tokenization, or convolution; attention is computed over raw pixel vectors.
> 3. **No multi-head attention, MLP, or full-rank projection layers**: the attention matrices $W^Q, W^K, W^V$ are diagonal, and the Transformer uses only a single head and 6 layers.
>
> By contrast, state-of-the-art models employ deep, multi-head Transformers or UNet-style CNNs with convolutional inductive bias, positional encodings, and patch-based input representations that are specifically designed to optimize perceptual metrics such as FID.
>
> Our focus in this paper is on **mechanistically understanding the denoising capabilities of the attention mechanism in isolation**. RMSE serves as the most direct measure of denoising accuracy in this setting. Nevertheless, we agree that FID is a useful complementary signal, and we view incorporating more perceptual metrics into future, more expressive variants of our architecture as an exciting future direction.
>
> ### CIFAR10/CIFAR100 FID-vs-Layer
> | Layer | CIFAR-10 (Theory) | CIFAR-10 (Trained) | CIFAR-100 (Theory) | CIFAR-100 (Trained) |
> | ----- | ----------------- | ------------------ | ------------------ | ------------------- |
> | 1     | 405               | 343.97 ± 1.36      | 392                | 324.51 ± 2.15       |
> | 2     | 325               | 195.97 ± 1.86      | 301                | 179.15 ± 1.44       |
> | 3     | 238               | 98.03 ± 1.00       | 217                | 95.07 ± 0.91        |
> | 4     | 167               | 43.26 ± 0.77       | 155                | 46.24 ± 1.06        |
> | 5     | 113               | 21.87 ± 0.54       | 108                | 25.25 ± 0.69        |
> | 6     | 78.3              | 19.31 ± 0.15       | 76.6               | 22.88 ± 0.38        |

---

> > ### Comment · Reviewer_YpJ6 · 2025-08-04
> >
> > Thanks for the rebuttal that addresses my concerns.

---

> > > ### Author Response · Authors · 2025-08-07
> > >
> > > We thank the reviewer again for their helpful discussion and suggestions.

---

### Comment · Area_Chair_DoCy · 2025-08-04
**Please carefully read the rebuttal and start the discussion**

Dear Reviewers and Authors,

Thank you all for your efforts so far. As the author–reviewer discussion period will conclude on **August 6**, please start the discussion as soon as possible.


**For Reviewers:**
Please read the authors’ responses and, if necessary, continue the discussion with them.

* If your concerns have been addressed, consider updating your review and score accordingly.

* If some concerns remain, or if you share concerns raised by other reviewers, clearly state these in your review and consider adjusting your review (positively or negatively).

* If you feel that your concerns have not been addressed, you may also choose to keep your review as is.

* I will follow up with you again during the reviewer–AC discussion period (August 7–13) to finalize the reviews and scores.


**For Authors:**
If you have not already done so, please respond to all questions raised by the reviewers. Keep your responses factual, concise, and ensure that every point raised is addressed.

Best regards,

The AC

---

### Note · Authors · 2025-08-12

We thank the reviewers and AC for the thoughtful discussions.

# Main strengths & contributions

* **Mechanistic understanding of Transformers as denoisers**: We give a **simple attention-based construction** that implements a **kernel-weighted update** (KWU) step per layer, **unifying manifold denoising and (in-context) diffusion denoising**.
  - Xi3K: *“**Novel interpretation** of transformer computation as a denoiser”*
  - eksT: *“Well-motivated… **solid theoretical derivations**… supported by experiments.”*
  - YpJ6: *"a **unified mathematical perspective**… novel and well-substantiated."*

* We characterize how **trained Transformers learn to implement more efficient algorithms than standard denoising,** via learnable witness tokens and anisotropic diffusion.
    - YpJ6: *"Learnable witness tokens... yield significant efficiency gains and opening new avenues for Transformer-based generative modeling."*


* We **empirically validate** our theory on MNIST and CIFAR-10, and CIFAR-100 (added in rebuttal).
    - YpJ6: *"experiments...confirm the theoretical claims in practice."*
    - Xi3K: *"This theoretical claim is further supported by empirical results on MNIST and CIFAR-10..."*



# Key criticisms & our responses

- **Experiments are on small image datasets**. Lack **standard metrics like FID**. (YpJ6, eksT, Xi3K)

  &rarr; Response: We provide additional experiments on **CIFAR100**, and provide **FID evaluations**. We plan additional perceptual metrics and datasets.

- **Model scalability** concerns. **No comparison with mainstream methods** (e.g. DiT).  (eksT, YpJ6)

  &rarr; Response: Our simple architecture helps **expose how attention layers implement denoising**. We acknowledge this concern, and plan to include more complex baselines in the future.

- **Clarification on motivation and setup** (ukLB)

  &rarr; Response: We clarify that our problem setup is consistent with standard diffusion formulations, and emphasize our theory as a **simple, interpretable attention mechanism**, rather than a universal-approximation result.

- Other questions were raised and addressed in individual rebuttals.

- **Edits to next draft**:
    - **Clarify motivation, setup and presentation**, as suggested by each reviewer.
    - **Discuss practical applications and extensions** (ukLB, eksT): including (1) adaptive generation via recurrent layers, (2) conditional generation via cross-attention and (3) efficient/interpretable anisotropic denoising.

---

### Decision · Program_Chairs · 2025-09-17

**Decision:**

Accept (poster)

**Comment:**

This paper develops a unified theoretical framework showing that Transformers can implement various diffusion-based denoising algorithms, supported by the novel idea of witness tokens for efficient score approximation and empirical validation on MNIST, CIFAR-10, and CIFAR-100. Reviewers found the work well-motivated, rigorous, and conceptually novel, highlighting its mechanistic interpretation of attention as a denoiser and its potential implications for generative modeling.

The main concerns were the limited experimental scope, absence of large-scale evaluations, and lack of comparisons with stronger baselines. The rebuttal partially addressed these by adding CIFAR-100 results, FID evaluations, and clarifying the setup, while acknowledging the need for broader benchmarks in future work.

Overall, the contribution is seen as original and impactful, providing valuable theoretical insights despite modest empirical scale.